# Longitudinal analysis of invariant natural killer T cell activation reveals a cMAF-associated transcriptional state of NKT10 cells

Harry Kane[1], Nelson M LaMarche[2], Áine Ní Scannail[3], Amanda E Garza[3], Hui-Fern Koay[3,4], Adiba I Azad[3], Britta Kunkemoeller[3], Brenneth Stevens[1,3], Michael B Brenner[2], Lydia Lynch[1,3]*

[1]Trinity Biomedical Science Institute, Trinity College Dublin, Dublin, Ireland; [2]Division of Rheumatology, Inflammation, and Immunity, Brigham and Women's Hospital, Harvard Medical School, Boston, United States; [3]Division of Endocrinology, Diabetes, and Hypertension, Brigham and Women's Hospital, Harvard Medical School, Boston, United States; [4]Department of Microbiology and Immunology, Peter Doherty Institute for Infection and Immunity, University of Melbourne, Melbourne, Australia

**Abstract** Innate T cells, including CD1d-restricted invariant natural killer T (iNKT) cells, are characterized by their rapid activation in response to non-peptide antigens, such as lipids. While the transcriptional profiles of naive, effector, and memory adaptive T cells have been well studied, less is known about the transcriptional regulation of different iNKT cell activation states. Here, using single-cell RNA-sequencing, we performed longitudinal profiling of activated murine iNKT cells, generating a transcriptomic atlas of iNKT cell activation states. We found that transcriptional signatures of activation are highly conserved among heterogeneous iNKT cell populations, including NKT1, NKT2, and NKT17 subsets, and human iNKT cells. Strikingly, we found that regulatory iNKT cells, such as adipose iNKT cells, undergo blunted activation and display constitutive enrichment of memory-like cMAF[+] and KLRG1[+] populations. Moreover, we identify a conserved cMAF-associated transcriptional network among NKT10 cells, providing novel insights into the biology of regulatory and antigen-experienced iNKT cells.

*For correspondence: llynch@bwh.harvard.edu

Competing interest: The authors declare that no competing interests exist.

## Editor's evaluation

This study presents a valuable finding on transcriptional profiles of various subsets of activated invariant natural killer T (iNKT) cells using longitudinal scRNA-seq analysis. The evidence supporting the conclusions is solid with rigorous and thorough bioinformatic analyses. The work will be of interest to scientists within the field of iNKT cells.

## Introduction

Activation of T cells following recognition of cognate antigen is essential for mounting effective immune responses against pathogens and tumors (*Kumar et al., 2018*). Typically, in the case of MHC-restricted adaptive CD4[+] and CD8[+] T cells, this requires extensive transcriptional remodeling over several days to facilitate proliferation and differentiation of naive T cells into clonal effector populations that traffic to sites of infection or tissue damage (*Chen et al., 2018a*). Transcriptional and metabolic remodeling is also needed to generate memory T cells that can be rapidly reactivated following

secondary antigen encounter during reinfection (*Chen et al., 2018a*). Innate T cells, including CD1d-restricted invariant natural killer T (iNKT) cells, contrast and complement this paradigm by exiting thymic development as poised 'effector-memory-like' cells already capable of mounting potent cytokine responses within minutes of activation. This allows iNKT cells to rapidly transactivate other immune populations and orchestrate immune responses (*Smyth et al., 2005*; *Reilly et al., 2012*). Activation also induces iNKT cell proliferation, generating an expanded pool of effector cells within 72 hr, most of which subsequently undergo apoptosis as the expanded iNKT cell pool contracts within 7 days (*Cameron and Godfrey, 2018*; *Wilson et al., 2003*; *Parekh et al., 2005*). However, some iNKT cells persist after the immune response subsides (*Wilson et al., 2003*; *Parekh et al., 2005*; *Shimizu et al., 2014*), and there is evidence that antigen challenge induces long-term changes in the iNKT cell repertoire analogous to memory T cell differentiation. For example, several studies have demonstrated that activation of iNKT cells with α-galalctosylceramide (αGalCer), a potent glycolipid antigen, induces the emergence of novel KLRG1$^+$ and Follicular Helper iNKT (NKT$_{FH}$) cell populations that are greatly enriched after 3–7 days, and still detectable >30 days after αGalCer challenge (*Shimizu et al., 2014*; *Chang et al., 2011*; *Murray et al., 2021*; *Chen et al., 2018b*). However, our knowledge of the transcriptional programs underpinning iNKT cell activation remains limited, and there are also relatively few transcriptional resources available for studying activated iNKT cells, especially compared to adaptive T cells (*Andreatta et al., 2021*).

Analysis and interpretation of iNKT cell biology is also challenging because iNKT cells exhibit heterogeneity, including NKT1, NKT2, and NKT17 subsets that broadly mirror CD4$^+$ Th1, Th2, and Th17 cells (*Engel et al., 2016*). Past studies of NKT1, NKT2, and NKT17 subsets largely focused on iNKT cell thymic development or steady-state phenotype in the absence of activation (*Engel et al., 2016*; *Harsha Krovi et al., 2020*; *Lee et al., 2015*; *Baranek et al., 2020*), and less is known about iNKT cell subsets after activation. Using parabiosis models, we and others have also shown that iNKT cells are predominantly tissue resident (*Lynch et al., 2015*; *Thomas et al., 2011*), and that this can strongly influence their biology (*LaMarche et al., 2020*). For example, iNKT cells resident in adipose tissue exhibit an unusual regulatory phenotype characterized by increased KLRG1 expression, reduced expression of the transcription factor promyelocytic leukemia zinc-finger (PLZF), and increased production of IL-10 through an IRE1a-XBP1s-E4BP4 axis, enabling these cells to suppress inflammation and promote metabolic homeostasis (*Lynch et al., 2015*; *LaMarche et al., 2020*). Interestingly, *Sag et al., 2014* demonstrated that IL-10$^+$ iNKT (NKT10) cells emerge in other organs such as the spleen after repeated antigen challenge (*Sag et al., 2014*), indicating that TCR stimulation can induce a regulatory phenotype, and that NKT10 cells can potentially be considered a memory-like population. However, the relationship between NKT10 cells and other memory-like populations, such as KLRG1$^+$ and NKTFH cells, remains unclear. Furthermore, it is unknown whether similar factors regulate NKT10 cells present in adipose tissue versus those induced after antigen challenge.

To characterize transcriptional remodeling in activated iNKT cells while also considering subset and tissue-associated heterogeneity, we performed single-cell RNA-sequencing (scRNA-seq) of 48,813 murine iNKT cells from spleen and adipose tissue at steady state and 4 hr, 72 hr, and 4 weeks after in vivo stimulation with αGalCer, as well as after repeated αGalCer challenge. We also reanalyzed published human and murine data to generate a transcriptomic atlas of iNKT cell activation states. We found that activation induces rapid and extensive transcriptional remodeling in iNKT cells, and that a common transcriptional framework underpins the activation of diverse iNKT cell populations. However, regulatory iNKT cell populations demonstrate largely blunted activation in response to αGalCer and display enrichment of memory-like KLRG1$^+$ and cMAF$^+$ iNKT cell subsets expressing a T regulatory type 1 (Tr1) cell gene signature. We also show that cMAF$^+$ iNKT cells are enriched for NKT10 cells and express a gene signature similar to NKT$_{FH}$ cells. Overall, this study provides novel insights into longitudinal transcriptional remodeling in activated iNKT cells and the phenotype of regulatory iNKT cells, while also generating a novel transcriptomic resource for interrogation of iNKT cell biology.

## Results

### iNKT cells undergo rapid and extensive transcriptional remodeling in response to αGalCer

To investigate transcriptional remodeling in activated iNKT cells, we performed 10× scRNA-seq of whole murine adipose and splenic iNKT cells 4 hr, 72 hr, and 4 weeks after in vivo stimulation with αGalCer and reanalyzed our published scRNA-seq of steady-state murine adipose and splenic iNKT cells (*LaMarche et al., 2020*; GSE142845, *Figure 1A*). We first analyzed our steady-state, 4 hr and 72 hr splenic iNKT cell data. After quality control measures, we obtained 16,701 splenic iNKT cells, including >4000 cells per activation state. After performing uniform manifold approximation and projection (UMAP), we observed minimal overlap between iNKT cells from different activation states (*Figure 1B*), indicating that iNKT cells undergo rapid and extensive transcriptional remodeling during early activation. Using gene expression analysis (*Supplementary file 1*), we found that steady-state iNKT cells displayed enrichment of NKT1 and NKT17 cell markers such as *Il2rb*, *Klrb1c*, *Rorc*, and *Il7r* (*Figure 1C*; *Engel et al., 2016*), but following activation iNKT cells rapidly downregulated these genes within 4 hr and upregulated expression of T cell activation markers and cytokines, including *Il2ra*, *Irf4*, *Nr4a1*, *Pdcd1*, *Ifng*, *Il4,* and *Il17a* (*Figure 1C*). This was accompanied by increased expression of *Zbtb16* (PLZF) and the PLZF regulon genes *Icos* and *Cd40lg* (*Figure 1C*), consistent with published data demonstrating that PLZF is required for the innate response of iNKT cells to antigen and is upregulated after activation (*Kovalovsky et al., 2008*; *Oleinika et al., 2018*). Activated iNKT cells also downregulated expression of the transcription factor *Id2* (*Figure 1C*), which plays an essential role in normal iNKT cell activation (*Stradner et al., 2016*), and upregulated expression of genes regulating T cell metabolic activation, including *Myc*, *Hif1a*, and *Tfrc* (*Marchingo et al., 2020*; *Finlay et al., 2012*; *Wang et al., 2018c*), suggesting that activated iNKT cells undergo metabolic remodeling.

By 72 hr, however, expression of activation and cytokine genes was greatly reduced, and we identified enrichment of genes associated with proliferation, stem-like T cells, and NKT2 or stage 2 iNKT cells, including *Mki67*, *Slamf6*, *Tcf7*, and *Ccr7* (*Figure 1C*, *Figure 1—figure supplements 1 and 2*; *Utzschneider et al., 2016*; *Cohen et al., 2013*). We observed that some 72 hr cells displayed enrichment of T$_{FH}$ and NKT$_{FH}$ markers, including *Cxcr5*, *Il21*, and *Maf* (*Chang et al., 2011*; *Chen et al., 2018b*; *Andreatta et al., 2021*; *Andris et al., 2017*), and memory-like iNKT cell markers, such as *Itga4* and *Klrg1* (*Shimizu et al., 2014*; *Figure 1C*, *Figure 1—figure supplements 2 and 3*), corresponding with previous studies documenting the appearance of NKT$_{FH}$ and KLRG1$^+$ iNKT cells after αGalCer challenge (*Shimizu et al., 2014*; *Chang et al., 2011*; *Murray et al., 2021*; *Rampuria and Lang, 2015*). We also found increased expression of genes associated with the KLF2 regulon, including *Klf2* and *S1pr1* (*Figure 1C*). KLF2 is known to induce T cell thymic egress and trafficking through secondary lymphoid organs (*Carlson et al., 2006*), and has been found to play an important role in iNKT cell migration and thymic emigration (*Harsha Krovi et al., 2020*; *Baranek et al., 2020*; *Wang et al., 2022*; *Wang and Hogquist, 2018a*). While iNKT cells are generally tissue resident under steady-state conditions (*Lynch et al., 2015*; *Thomas et al., 2011*), increased expression of *Klf2* and *S1pr1* at 72 hr post-αGalCer could suggest that activated iNKT cells may traffic to other sites. However, previous work has also shown that hepatic iNKT cells arrest after becoming activated (*Geissmann et al., 2005*; *Liew et al., 2017*). Therefore, further analysis of migration in iNKT cells from different organs and at distinct stages of activation will be required to elucidate if and when activated iNKT cells undergo migration.

Having identified enrichment of *Myc*, *Hif1a* and *Tfrc* 4 hr post-αGalCer, we wondered what type of metabolic remodeling activated iNKT cells undergo in vivo. To map metabolic gene changes during iNKT cell activation, we generated gene module scores using the KEGG pathway (*Kanehisa et al., 2021*) and Gene Ontology Consortium (*Carbon, 2021*) databases (*Supplementary file 2*) and scored our data. We found 4 hr activated cells upregulated glycolysis, amino acid metabolism, polyamine synthesis, and fatty acid synthesis signatures, whereas oxidative signatures were downregulated compared to steady-state iNKT cells (*Figure 1D*). This suggests that, despite being poised at steady state for cytokine production, activated iNKT cells, like adaptive T cells, may switch on aerobic glycolysis and upregulate biosynthethic pathways to fuel cytokine production, growth, and proliferation (*Marchingo et al., 2020*; *Angiari et al., 2020*; *Wu et al., 2020*). Our data is also consistent with recent work identifying glucose as an important fuel for iNKT cell effector function (*Fu et al., 2019*; *Kumar et al., 2019*). Interestingly, we found that 72 hr activated cells engage oxidative signatures while maintaining elevated expression of glycolytic genes (*Figure 1D*), suggesting that

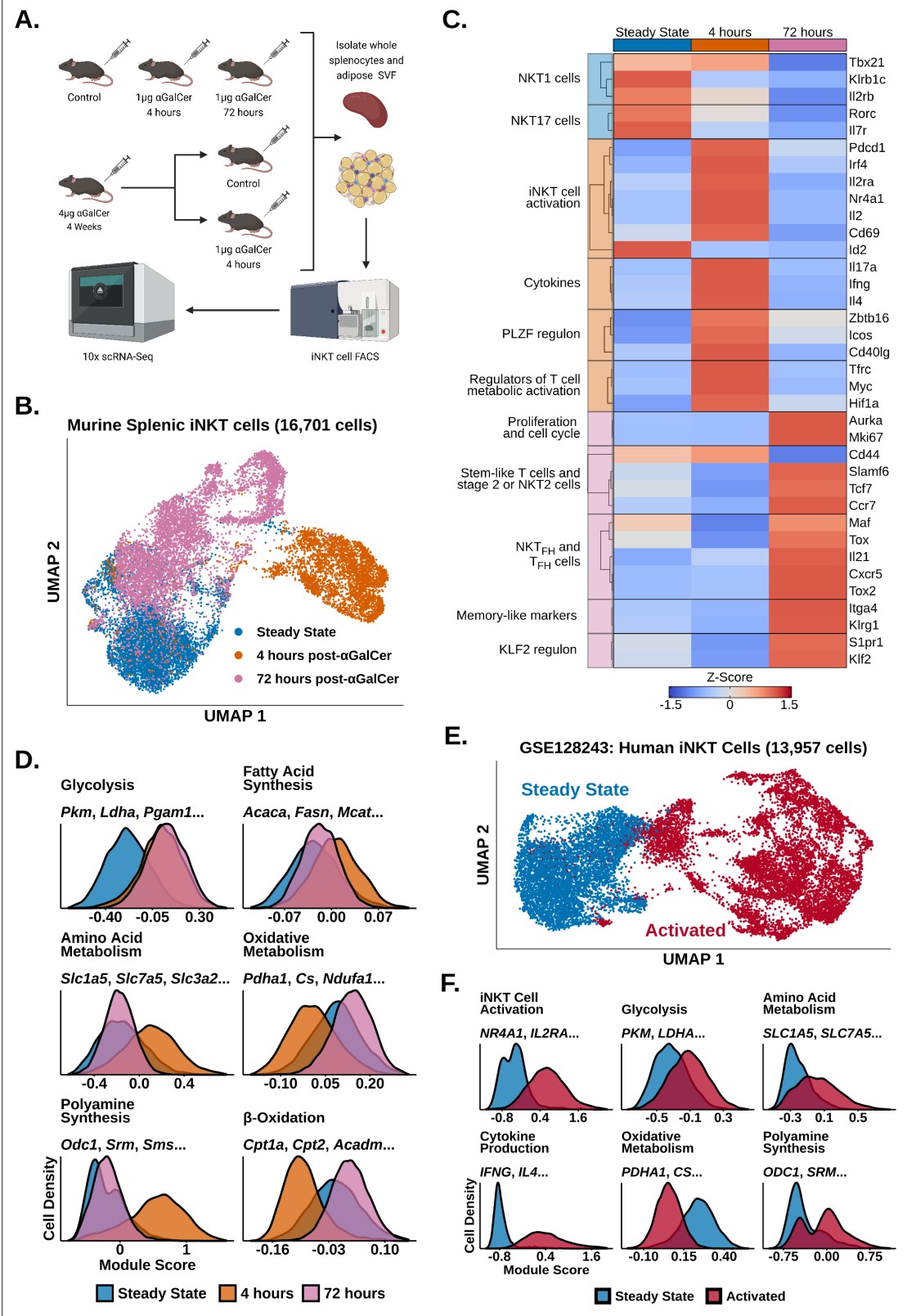

**Figure 1.** Invariant natural killer T (iNKT) cells undergo rapid and extensive transcriptional remodeling in response to α-galalctosylceramide (αGalCer). (**A**) Cartoon illustrating the experimental design for the generation of all scRNA-seq data. (**B**) Uniform manifold approximation and projection (UMAP) of murine splenic iNKT cells with cell cycle regression. (**C**) Heatmap of scaled averaged gene expression with hierarchical clustering in the data from (**B**). (**D**)

*Figure 1 continued on next page*

*Figure 1 continued*

Histograms showing expression of metabolic gene module scores in the data from (**B**). (**E**) UMAP of human peripheral blood mononuclear cell (PBMC) iNKT cells reanalyzed from GSE128243. (**F**) Histograms showing expression of functional and metabolic gene module scores in the data from (**E**).

The online version of this article includes the following figure supplement(s) for figure 1:

**Figure supplement 1.** Single cell gene expression of splenic iNKT cells at steady state, and 4 and 72 hours post activation.

**Figure supplement 2.** Expression plots of proliferation gene scores and select genes in splenic iNKT cells at 72 hours post activation.

**Figure supplement 3.** Enrichment of cMAF in activated iNKT cells.

the metabolic requirements of iNKT cells change across different activation states. We also observed reduced expression of polyamine synthesis and amino acid metabolism signatures 72 hr post-αGalCer, indicating that those pathways may be coupled to early iNKT cell activation and cytokine production, while oxidative metabolism is associated with proliferation when the iNKT cell pool expands four- to tenfold in vivo by 72 hr ; (*Wilson et al., 2003*; *Parekh et al., 2005*).

Having profiled transcriptional remodeling in activated murine iNKT cells, we wondered whether similar remodeling occurs in human iNKT cells. To investigate human iNKT cell activation, we reanalyzed published scRNA-seq data of human iNKT cells isolated from peripheral blood mononuclear cells (PBMCs) and stimulated ex vivo with phorbol 12-myristate 13-acetate (PMA) and ionomycin (GSE128243; *Zhou et al., 2020*). Following quality control measures, we obtained 13,957 cells and found that human iNKT cells also undergo rapid and extensive transcriptional remodeling after activation (*Figure 1E and F*). Furthermore, activated human iNKT cells recapitulated the metabolic gene reprogramming observed in activated murine iNKT cells, displaying upregulated glycolytic, amino acid metabolism and polyamine synthesis signatures, and reduced expression of oxidative signatures (*Figure 1F*). Thus, transcriptional signatures of iNKT cell activation are conserved across species.

## Oxidative phosphorylation differentiates functional responses to αGalCer in NKT2 and NKT17 cells versus NKT1 cells

We next asked whether iNKT cell subsets expressed different transcriptional signatures after activation. We performed subclustering of murine splenic iNKT cells at steady state and 4 hr post-αGalCer, and identified clusters corresponding to NKT1, NKT2, and NKT17 cells (*Figure 2A*, *Figure 2—figure supplements 1 and 2*; see 'Methods') using the published transcription factors *Tbx21*, *Zbtb16*, and *Rorc,* and the flagship cytokines *Ifng*, *Il4*, *Il13*, *Il17a*, and *Il17f* (*Cameron and Godfrey, 2018*; *Engel et al., 2016*; *Venken et al., 2019*; *Figure 2B*, *Figure 2—figure supplements 1 and 2*). Notably, we found that all subsets upregulated *Zbtb16* after activation (*Figure 2B*), suggesting that PLZF may play a subset-independent role during activation. We found that *Tbx21* expression was nonspecifically increased across all iNKT cell subsets after activation (*Figure 2B*), and therefore, we did not use *Tbx21* to demarcate activated iNKT cells. When we performed gene expression analysis, we found that all subsets demonstrated upregulation of activation, cytokine, glycolysis, amino acid metabolism, polyamine synthesis, and fatty acid synthesis signatures after αGalCer (*Figure 2C*, *Figure 2—figure supplement 3*), indicating that a common transcriptional framework underpins the activation of functionally diverse iNKT cell subsets. We also identified genes specifically enriched in one or more subsets, such as *Gzmb* and *Ccl4* in NKT1 cells (*Figure 2C*; *Supplementary file 3*). Strikingly, we found that NKT2 and NKT17 cells, but not NKT1 cells, shared expression of many genes, including *Lif*, *Lta*, *Cd274,* and *Ncoa7* (*Figure 2C*). Activated NKT2 and NKT17 cells also demonstrated increased whole-transcriptome correlation compared to activated NKT1 cells (*Figure 2D*), indicating that activated NKT2 and NKT17 cells are transcriptionally similar compared to NKT1 cells.

To investigate the shared transcriptional signatures of NKT2 and NKT17 cells, we performed gene set enrichment analysis (GSEA) (*Korotkevich et al., 2016*; *Subramanian et al., 2005*) comparing activated NKT2 and NKT17 cells versus activated NKT1 cells using the KEGG pathway database (*Kanehisa et al., 2021*). We identified enrichment of oxidative phosphorylation (*Figure 2E*), suggesting that NKT2 and NKT17 cells use oxidative metabolism more than NKT1 cells. To validate this result, we first measured mitochondrial mass and membrane potential in thymic CD44+ NKT1, NKT2, and NKT17 cells from BALB/c mice (*Figure 2—figure supplement 4*; *Cameron and Godfrey, 2018*), and we found that NKT2 and NKT17 cells had significantly increased mitochondrial mass and membrane potential compared to NKT1 cells (*Figure 2F and G*). We also found that the expression of NK1.1, a

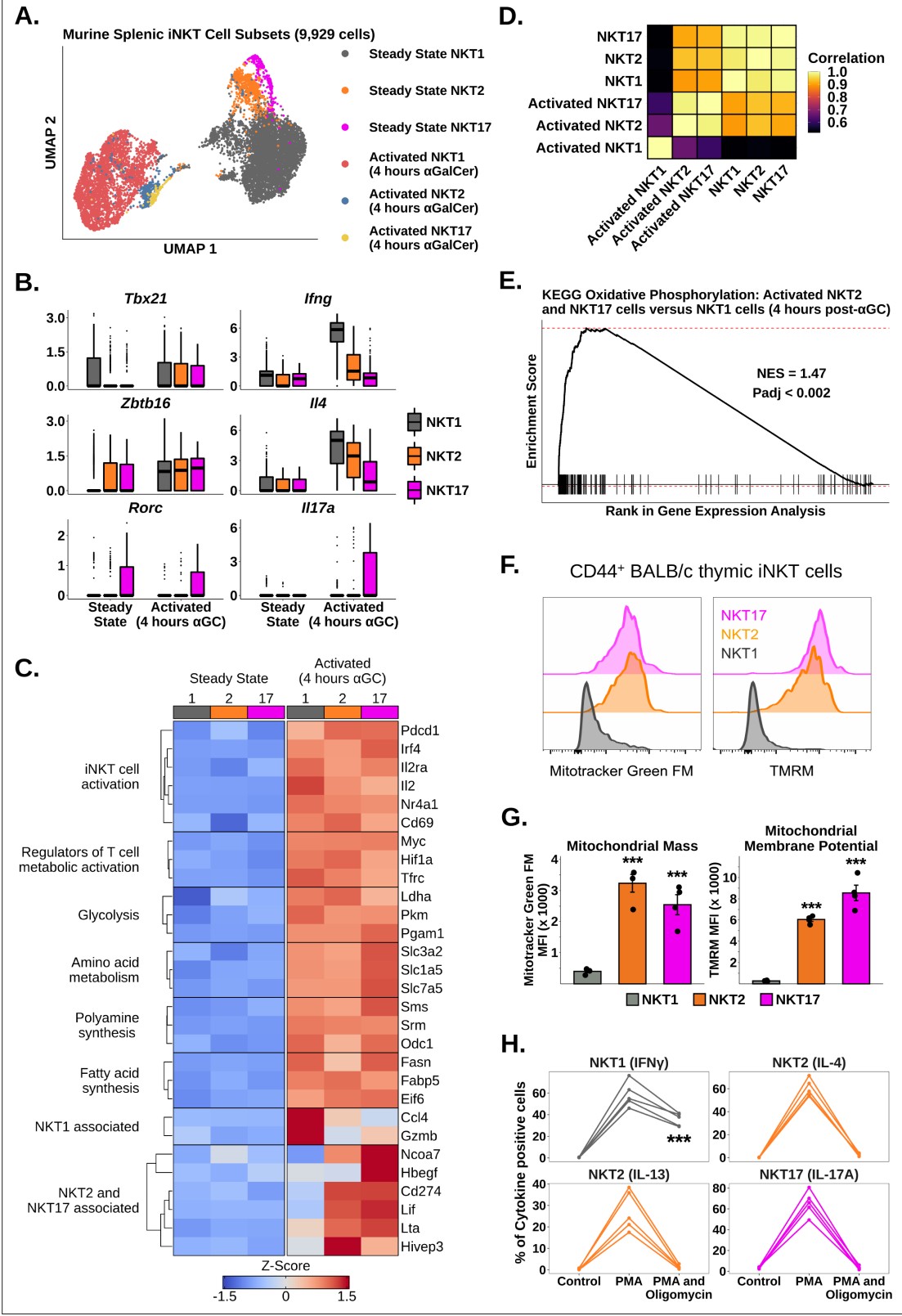

**Figure 2.** Oxidative phosphorylation differentiates functional responses to α-galalctosylceramide (αGalCer) in NKT2 and NKT17 cells versus NKT1 cells. (**A**) Uniform manifold approximation and projection (UMAP) of murine splenic invariant natural killer T (iNKT) cells. (**B**) Box plots showing gene expression in the data from (**A**). αGC denotes αGalCer. The central box plot horizontal line denotes the median value. (**C**) Heatmap of scaled averaged gene expression with hierarchical clustering in NKT1 (1), NKT2 (2), and NKT17 cells (17) from the data in (**A**). (**D**) Correlation plot of total normalized

*Figure 2 continued*

RNA counts for all genes across steady state and activated NKT1, NKT2, and NKT17 cell subsets. (**E**) Gene set enrichment analysis (GSEA) plot of KEGG oxidative phosphorylation comparing activated NKT2 cells and NKT17 cells versus activated NKT1 cells. NES, normalized enrichment score; Padj, adjusted p-value for the enrichment. (**F**) Histograms showing staining of MitoTracker Green FM and TMRM in CD44+ NKT1, NKT2, and NKT17 cells from BALB/c mouse thymus. iNKT cells were defined as live, single CD19- CD8- CD3low CD1d-PBS57 tetramer+ cells. Histograms were normalized to the mode. (**G**) Bar plots quantifying the mean fluorescence intensity (MFI) data from (**F**). N = 4 biological replicates from one experiment. Experiment performed at least twice. Student's unpaired *t*-test. Asterisks denote significance, * Padj<0.05; ** Padj<0.01; *** Padj<0.001. Source data provided in *Figure 2—source data 1*. (**H**) Line plots showing production of flagship cytokines by CD44+ BALB/c mouse thymic iNKT cells: NKT1 (IFNγ), NKT2 (IL-4 and IL-13), and NKT17 cells (IL-17A) without stimulation and after 50 ng phorbol 12-myristate 13-acetate (PMA) and 1 µg ionomycin in the absence or presence of 40 nM oligomycin (all 4 hr ex vivo). N = 5 biological replicates from one experiment. Experiment performed at least twice. One-way ANOVA and Tukey's post hoc test. Asterisks denote significance, * Padj<0.05; ** Padj<0.01; *** Padj<0.001. Source data provided in *Figure 2—source data 2*.

The online version of this article includes the following source data and figure supplement(s) for figure 2:

**Source data 1.** MFI expression levels of mitotracker and TMRM in iNKT cell subsets by flow cytometry.

**Source data 2.** Amount of cytokines produced in each iNKT cell subset with or without metabolic inhibitors.

**Figure supplement 1.** Distinct gene expression differences in iNKT cell subsets.

**Figure supplement 2.** UMAP plots of NKT subsets, resting and post-activation.

**Figure supplement 3.** Module score analysis of scRNA-Seq data reveals changes in metabolic pathway gene expression following iNKT cell activation.

**Figure supplement 4.** Flow cytometry validation of iNKT cell subset identity in the BALB/c CD43-HG/ICOS model using expression of flagship NKT2, NKT17 and NKT1 cell transcription factors.

**Figure supplement 5.** Correlation of mitochondrial marker expression with NKT subsets.

surface marker that is known to segregate NKT1 versus NKT2 and NKT17 cells (*Engel et al., 2016*; *Lee et al., 2013*), was able to distinguish distinct mitochondrial phenotypes among splenic iNKT cells from C57BL/6 mice (*Figure 2—figure supplement 5*). We next investigated whether NKT2 and NKT17 cells were more functionally dependent on oxidative metabolism than NKT1 cells by stimulating BALB/c thymic iNKT cells ex vivo with PMA and ionomycin for 4 hr in the presence or absence of oligomycin, to inhibit oxidative phosphorylation (*Lopes et al., 2021*). Treatment with oligomycin globally reduced cytokine production across all iNKT cell subsets; however, we found that production of IL-4, IL-13, and IL-17A was almost completely ablated compared to production of IFNγ (*Figure 2H*). We also found that treatment with oligomycin resulted in significantly reduced production of IL-4 but not IFNγ by splenic iNKT cells from C57BL/6 mice (*Figure 2—figure supplement 5*). Collectively, our data demonstrate that oxidative metabolism is essential for the production of NKT2 and NKT17 cytokines but less so for NKT1 cytokines.

## Adipose iNKT cells display blunted and delayed activation after αGalCer, enrichment of Tr1 cell markers, and hallmarks of chronic endogenous activation

We and others have shown that adipose iNKT cells display an unusual regulatory phenotype characterized by E4BP4 (*Nfil3*) expression and enrichment of NKT10 cells (*Lynch et al., 2015*; *LaMarche et al., 2020*; *Sag et al., 2014*). Having characterized activated splenic iNKT cells, we next investigated activated adipose iNKT cells. Combined analysis of adipose and splenic iNKT cells at steady state, 4 hr post-αGalCer, and 72 hr post-αGalCer returned 28,561 cells, including 11,860 adipose iNKT cells, and >2900 cells per activation state. When we performed UMAP, we found that adipose and splenic iNKT cells displayed minimal overlap (*Figure 3A*), indicative of constitutive transcriptional differences between adipose and splenic iNKT cells. Cross-dataset differential gene expression analysis ('Methods') identified 971 genes enriched among all adipose iNKT cells regardless of activation status or subset, versus only 65 genes enriched among all splenic iNKT cells (*Supplementary file 4*), indicating that adipose but not splenic iNKT cells retain expression of many conserved genes during activation. When we profiled these conserved genes using over-representation analysis against the KEGG pathway database, we identified Ribosome as the sole enriched pathway among splenic iNKT cells, whereas adipose iNKT cells displayed enrichment of pathways related to uptake from the extracellular environment (endocytosis, regulation of actin cytoskeleton, FcγR-mediated phagocytosis), adhesion (focal adhesion, leukocyte transendothelial migration), cytotoxicity and cell death (NK cell-mediated cytotoxicity, apoptosis), cellular senescence, chemokine signaling, and TCR signaling (*Figure 3B*).

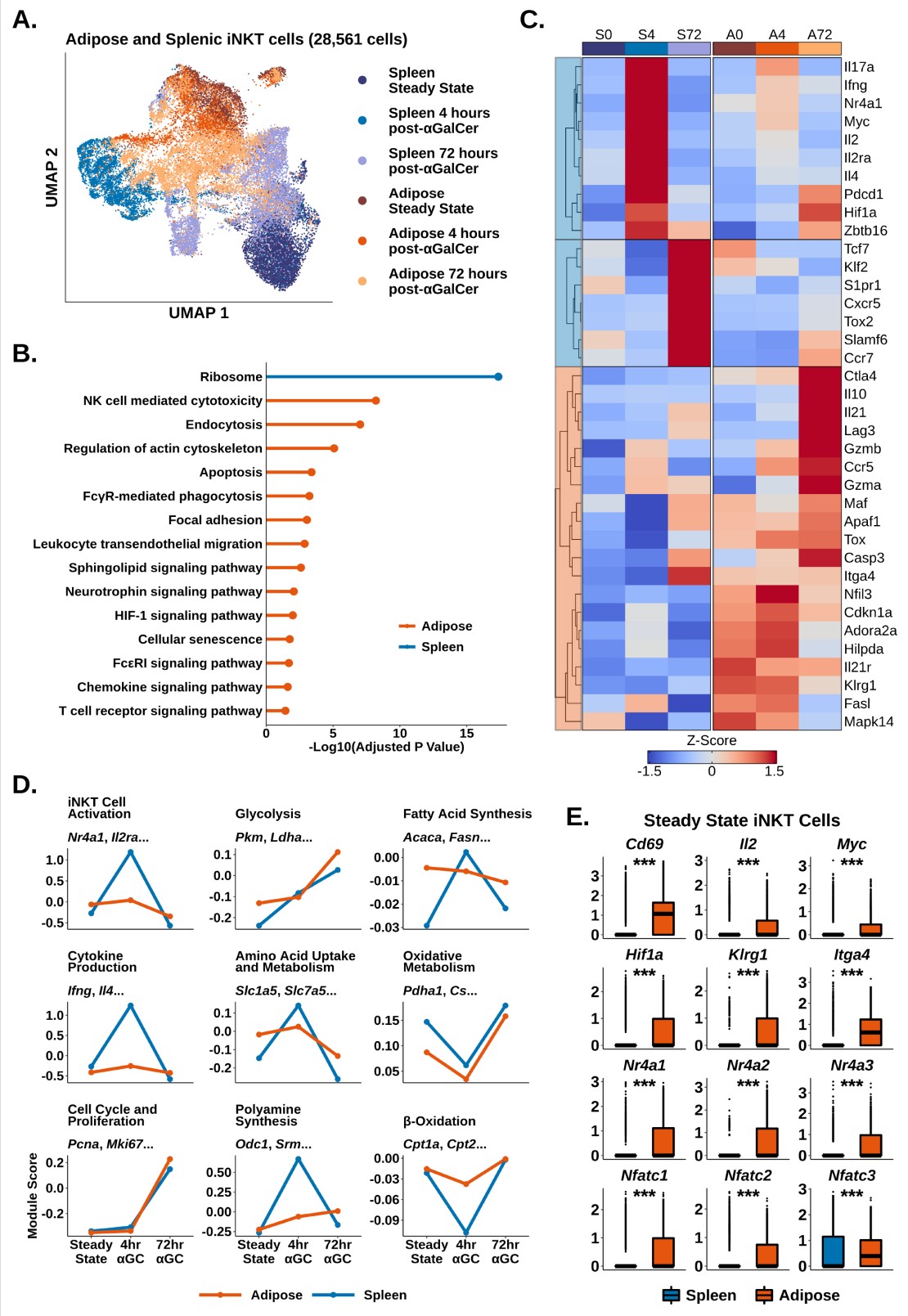

**Figure 3.** Adipose invariant natural killer T (iNKT) cells display blunted and delayed activation after α-galalctosylceramide (αGalCer), enrichment of Tr1 Cell markers, and hallmarks of chronic endogenous activation. (**A**) Uniform manifold approximation and projection (UMAP) of murine adipose and splenic iNKT cells with cell cycle regression. (**B**) Lollipop plot showing enrichment of nondisease KEGG pathways based on conserved enrichment 971 genes in murine adipose iNKT cells and 65 genes in murine splenic iNKT cells. Source data provided in *Figure 3—source data 1*. (**C**) Heatmap of scaled

*Figure 3 continued on next page*

*Figure 3 continued*

averaged gene expression with hierarchical clustering in the data from (**A**). (**D**) Line plots showing median expression of functional and metabolic gene module scores in the data from (**A**). (**E**) Box plots showing gene expression in murine steady-state splenic or adipose iNKT cells. The central box plot horizontal line denotes the median value.

The online version of this article includes the following source data and figure supplement(s) for figure 3:

**Source data 1.** Comparison of splenic and adipose iNKT cells at 72hr post-activation.

**Figure supplement 1.** Comparison of splenic and adipose iNKT cells at 72hr post-activation.

We next compared gene expression among individual iNKT cell activation states. Intriguingly, we found that there was greatly reduced expression of activation markers such as *Il2ra* and *Nr4a1* (Nur77) in adipose versus splenic iNKT cells at 4 hr post-αGalCer (*Figure 3C*). We previously found that Nur77 is enriched in adipose iNKT cells compared to splenic iNKT cells at steady state (*Lynch et al., 2015*; *LaMarche et al., 2020*), but these data suggest that adipose iNKT cells demonstrate reduced upregulation of Nur77 compared to splenic iNKT cells upon activation. Using module scoring, we found that adipose iNKT cells showed a blunted initial response to αGalCer characterized by reduced upregulation of activation and cytokine signatures, and reduced metabolic remodeling (*Figure 3D*). At 72 hr post-αGalCer, when proliferative gene signatures are upregulated, adipose and splenic iNKT cells showed similar enrichment of proliferation markers (*Figure 3D*), but adipose iNKT cells had reduced expression of stem-like markers such as *Tcf7* and *Slamf6*, and reduced expression of T_FH markers such as *Cxcr5* and *Tox2* (*Figure 3C*). Adipose iNKT cells showed increased expression of *Il10*, *Lag3*, *Ctla4*, *Il21*, *Ccr5*, *Hif1a*, *Maf*, and *Gzmb* (*Figure 3C*), which are markers typically associated with Tr1 cells, a heterogeneous population of regulatory T cells that do not express FOXP3 (*Yan et al., 2017*; *Gruarin et al., 2019*; *Chihara et al., 2016*; *Alfen et al., 2018*; *Mascanfroni et al., 2015*). We previously found that adipose iNKT cells do not express FOXP3 and instead express the transcription factor E4BP4, which regulates IL-10 production (*Lynch et al., 2015*; *LaMarche et al., 2020*). Other genes enriched among adipose iNKT cells included *Il21r*, the adenosine receptor *Adora2a*, and the exhaustion marker *Tox* (*Figure 3C*). In summary, activation with αGalCer induces differential transcriptional remodeling in adipose versus splenic iNKT cells, and the peak of the regulatory response in adipose iNKT cells is delayed compared to the rapid cytokine burst in splenic iNKT cells.

Notably, although adipose iNKT cells displayed blunted activation and metabolic remodeling compared to splenic iNKT cells, adipose iNKT cells were already enriched for activation, glycolysis, and amino acid metabolism gene signatures at steady state (*Figure 3D*), suggesting that adipose cells are activated at baseline. At steady state, adipose iNKT cells had increased expression of key activation markers such as *Il2*, *Cd69*, *Myc*, and *Nr4a* genes, as well as increased *Nfat* gene expression (*Figure 3E*). *Nr4a* and *Nfat* genes are commonly upregulated in chronically activated T cells, often in combination with *Tox* (*Liu et al., 2019*; *Seo et al., 2019*; *Kumar et al., 2020*), suggesting that adipose iNKT cells experience chronic activation. Furthermore, we found that steady-state adipose iNKT cells demonstrated increased expression of markers of antigen experience, *Klrg1* and *Itga4* (*Figure 3E*; *Shimizu et al., 2014*), and we have previously shown that KLRG1[+] iNKT cells are greatly enriched in adipose tissue (*Lynch et al., 2015*). We also found that adipose iNKT cells, but not splenic iNKT cells, continued to express cytokine transcripts at 72 hr post-αGalCer (*Figure 3—figure supplement 1*), indicative of a 'smoldering' activation phenotype among adipose iNKT cells. Overall, our data suggest that adipose iNKT cells experience chronic endogenous activation, which may dampen their ability to undergo further rapid activation in response to αGalCer, and could explain the nature of their regulatory phenotype.

## scRNA-seq identifies transcriptional signatures of adipose iNKT cell subset activation and adipose NKT10 cells

We have recently shown that adipose iNKT cells are more heterogeneous than originally anticipated. At steady state, these cells are comprised of NK1.1[+] NKT1 cells, NK1.1[-] NKT1 cells, and NKT17 cells (*LaMarche et al., 2020*). NK1.1[+] and NK1.1[-] adipose iNKT cells are functionally distinct as NK1.1[+] cells produce IFNγ, and NK1.1[-] iNKT cells produce IL-4 and IL-10. However, it is unknown whether these populations engage distinct molecular programs in response to αGalCer, a proposed therapy for type 2 diabetes and obesity. We also wondered whether the reduced

responsiveness to αGalCer in adipose iNKT cells was linked to any particular adipose iNKT cell population. To answer these questions, we performed analysis of adipose iNKT cells at steady state and 4 hr post-αGalCer. Unlike splenic iNKT cells, where activation accounted for most of the variance (*Figure 2A*), transcriptional differences between adipose NKT1 cells and NKT17 cells accounted for most of the variance among adipose iNKT cells (*Figure 4A*). We found that adipose NKT1 and NKT17 cells both upregulated expression of activation, cytokine, and metabolic gene signatures after αGalCer (*Figure 4B*). Comparison of adipose and splenic iNKT cell subsets showed that all adipose iNKT cell subsets had reduced activation and cytokine gene signature expression, and reduced metabolic remodeling at 4 hr post-αGalCer (*Figure 4C*). This indicates that all adipose iNKT cell subsets respond to αGalCer but no single subset (e.g., NKT1 or NKT17) was uniquely hyporesponsive vis-a-vis the spleen. Interestingly, we identified increased oxidative gene expression in adipose NKT17 cells versus adipose NKT1 cells (*Figure 4C*), similar to our finding in splenic iNKT cells (*Figure 2*). We have previously shown that γδ17 cells also display enrichment of oxidative metabolism (*Lopes et al., 2021*), suggesting that this is a conserved feature of innate T cells that produce IL-17.

Analysis of cytokine gene expression among adipose iNKT cells revealed that *Il10* was only expressed by NKT1 cells (*Figure 4B*). Since cytokine$^{pos}$ adipose NKT1 cells (cluster 6) lacked or had downregulated expression of *Klrb1c* (NK1.1) by 4 hr post-αGalCer (*Figure 4B*), we could not stratify cytokine production in adipose NKT1 cells using *Klrb1c*. Therefore, we performed unbiased fine clustering of cytokine$^{pos}$ adipose NKT1 cells. We identified one population of cells co-expressing *Ifng*, *Il4,* and *Il2*, and a second population enriched for *Il10* (*Figure 4D*), suggesting that adipose NKT10 cells are a distinct population. Analysis of cytokine$^{pos}$ adipose iNKT cells at 72 hr post-αGalCer, when expression of *Il10* was highest among adipose iNKT cells (*Figure 3*), identified some NKT10 cells co-expressing *Il10* with *Ifng*, *Il4,* and *Il21*, and other NKT10 cells that co-expressed *Il10*, *Ifng,* and *Gzmb*, suggesting that expanded NKT10 cells are heterogeneous (*Figure 4—figure supplement 1*). To investigate the transcriptional profile of adipose NKT10 cells, we performed global analysis of *Il10*$^{pos}$ versus *Il10*$^{neg}$ adipose iNKT cells at 4 hr and 72 hr post-αGalCer. We identified 207 genes enriched in adipose NKT10 cells (*Supplementary file 5*), including the Tr1 cell markers *Lag3*, *Ctla4*, *Pdcd1*, *Gzmb*, *Il21*, *Hif1a*, *Maf,* and *Ccr5* (*Figure 4E*; *Chihara et al., 2016*; *Mascanfroni et al., 2015*; *Grossman et al., 2004*; *Apetoh et al., 2010*), suggesting that adipose NKT10 cells may be functionally similar to Tr1 cells. In summary, our analysis suggests that adipose iNKT cells primarily segregate by function after αGalCer, and that regulatory adipose NKT10 cells are a transcriptionally distinct population similar to Tr1 cells.

## Chronic activation of splenic iNKT cells induces an adipose-like phenotype and the emergence of Tr1 iNKT cells

Since adipose iNKT cells displayed blunted and delayed activation after αGalCer, and enrichment of Tr1 cell markers, we wondered whether these were conserved features of regulatory iNKT cell biology. To answer this question, we repeatedly activated splenic iNKT cells, which induces IL-10 production (*Sag et al., 2014*). We sequenced 5433 αGalCer activated splenic iNKT cells, including 2117 cells isolated 4 weeks after mice received one dose of αGalCer (resting) and 3316 cells isolated 4 hr after reactivation with a second dose of αGalCer (reactivated; *Figure 5A*). Comparison of iNKT cells at steady state (no αGalCer) and resting (4 weeks post-αGalCer) revealed that resting iNKT cells displayed a reduced response to restimulation with αGalCer, similar to the adipose iNKT cell response to one dose of αGalCer (*Figure 5B*). This indicates that blunted activation after αGalCer is a conserved feature of regulatory iNKT cells. Furthermore, gene expression analysis revealed that resting iNKT cells were transcriptionally similar to adipose iNKT cells, displaying reduced *Ifng*, *Il4,* and *Il2* expression, and increased expression of Tr1 cells markers such as *Gzmb*, *Il10*, *Maf*, *Il21*, *Ctla4*, *Hif1a*, *Ccr5,* and *Lag3*, especially after αGalCer rechallenge (*Figure 5C*), suggesting that regulatory iNKT cells are similar to Tr1 cells. We also reanalyzed previously published microarray data of control and αGalCer-pretreated splenic iNKT cells (GSE47959; *Sag et al., 2014*; *Figure 5—figure supplement 1*) and identified a similar activation phenotype in αGalCer-pretreated splenic iNKT cells versus control iNKT cells after short-term αGalCer stimulation. Interestingly, although resting iNKT cells expressed *Tox* (*Figure 5C*), we did not identify enrichment of other chronic activation markers, such as *Nr4a1* (Nur77) (*Figure 5—figure supplement 2*, *Supplementary file 6*), indicating that prior exposure of splenic

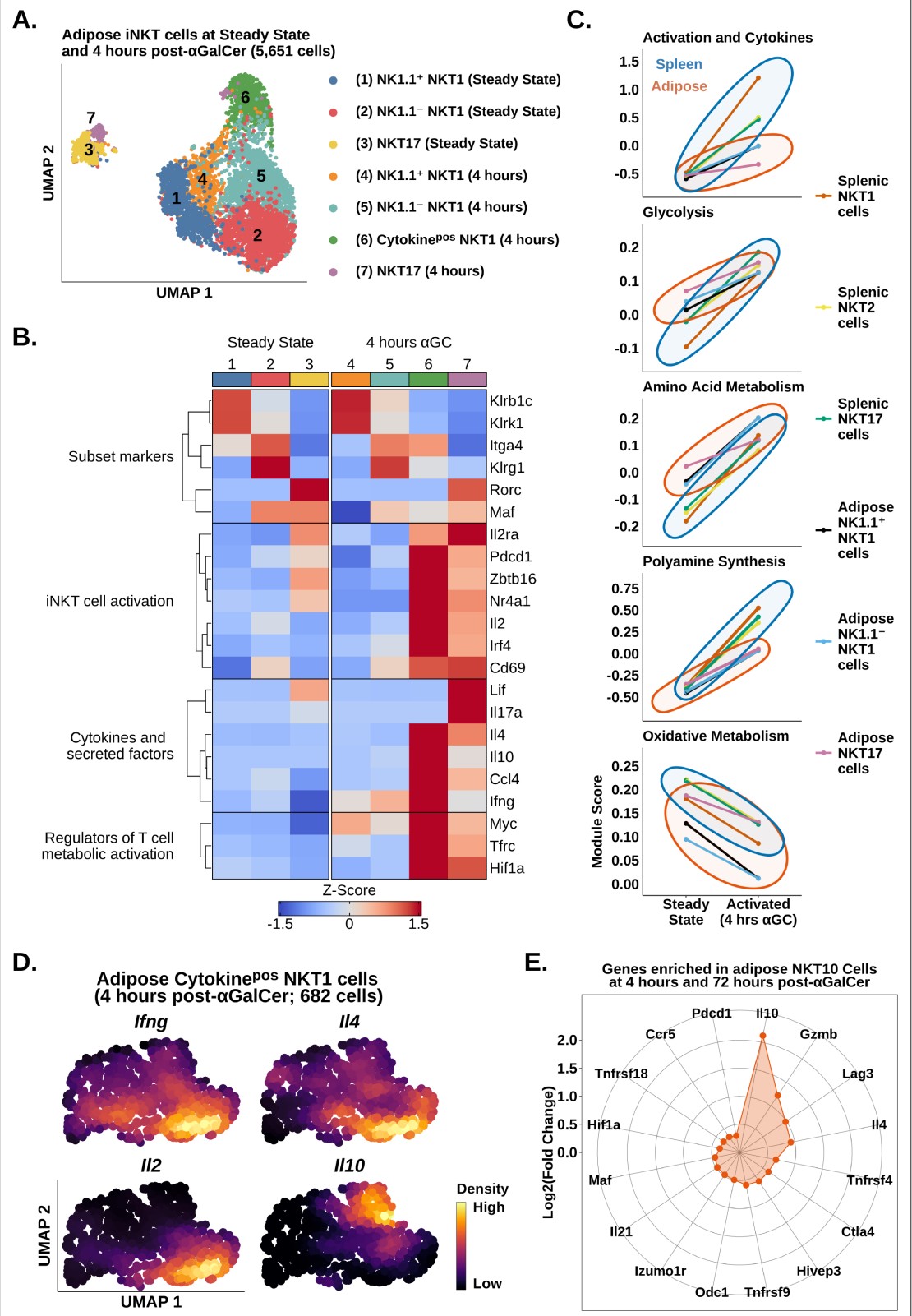

**Figure 4.** scRNA-seq identifies transcriptional signatures of adipose invariant natural killer T (iNKT) cell subset activation. (**A**) Uniform manifold approximation and projection (UMAP) of murine adipose iNKT cells. (**B**) Heatmap of scaled averaged gene expression with hierarchical clustering in the data from (**A**). (**C**) Line plots showing median expression of functional and metabolic gene module scores in murine adipose and splenic NKT1, NKT2, and NKT17 cell subsets. (**D**) Density plots of gene expression in murine adipose cytokine^pos NKT1 cells at 4 hr post-αGalCer (cluster 6, **A**). (**E**)

*Figure 4 continued on next page*

*Figure 4 continued*

Radar chart showing Log2(Fold Change) values of genes enriched in *Il10*^pos adipose iNKT cells versus *Il10*^neg adipose iNKT cells at 4 hr and 72 hr post-α-galalctosylceramide (post-αGalCer).

The online version of this article includes the following figure supplement(s) for figure 4:

**Figure supplement 1.** Adipose iNKT cell heterogeneity at 72hr post aGalCer activation.

iNKT cells to antigen does not completely reproduce the phenotype of adipose iNKT cells, which may be exposed to chronic endogenous activation in situ.

Since we had identified a distinct population of NKT10 cells in adipose tissue after αGalCer, we wondered whether we could also identify an NKT10 population among reactivated splenic iNKT cells. Graph-based clustering of reactivated splenic iNKT cells revealed two major populations, clusters A and B (***Figure 5D***), which were differentiated by their response to reactivation. Only cluster B cells (~50% of cells) demonstrated significant activation, cytokine transcript expression, and metabolic gene reprogramming after the second dose of αGalCer (***Figure 5E***, ***Figure 5—figure supplement 3***). Functional stratification of reactivated splenic iNKT cells revealed a population of NKT1 cells co-expressing *Ifng* and *Il4*, a population of NKT17 cells expressing *Il17a*, a population of NKT10 cells co-expressing *Il10* and *Maf*, as well as intermediate *Ifng* and *Il4*, and a population of cells co-expressing *Gzma* and the memory-like iNKT cell marker *Klrg1* (***Figure 5E***, ***Figure 5—figure supplement 3***). These data suggest that prior immunization with αGalCer is associated with the appearance of two new functional iNKT cell populations expressing *Klrg1* or *Maf*. We confirmed these findings using flow cytometry, identifying enrichment of Granzyme A⁺ KLRG1⁺ iNKT cells in the spleen at 4 weeks post-αGalCer versus steady state (***Figure 5F***), increased expression of cMAF at 4 weeks post-αGalCer (***Figure 5—figure supplement 4***), and increased IL-10 production among restimulated cMAF⁺ iNKT cells compared to cMAF⁻ iNKT cells at 4 weeks post-αGalCer (***Figure 5G***). We also found that expression of Granzyme A and IL-10 was mutually exclusive at the protein level (***Figure 5—figure supplement 4***), matching our scRNA-seq analysis. Therefore, these data suggest that repeated antigen exposure induces a regulatory phenotype in splenic iNKT cells, which is associated with the appearance of two new functionally distinct iNKT cell populations expressing cMAF and KLRG1.

## Memory-like cMAF⁺ and KLRG1⁺ iNKT cells are induced in the spleen following αGalCer challenge, and similar populations are constitutively present in adipose tissue

Having identified enrichment of iNKT cells expressing *Maf* and *Klrg1* among reactivated splenic iNKT cells, we wondered whether we could identify similar populations among resting iNKT cells. We also sought to characterize the transcriptional profile of these populations. Analysis of steady-state and resting iNKT cells revealed distinct NKT1, NKT2, NKT17, and cycling cell populations on the basis of graph-based clustering, spatial separation, and *Tbx21*, *Zbtb16*, *Rorc*, and *Mki67* expression (***Figure 6A and B Figure 2—figure supplement 1***, ***Figure 6—figure supplement 1***). We additionally found that NKT1 cells separated into two distinct clusters, NKT1-A and NKT1-B cells, with NKT1-B cells displaying increased expression of genes associated with the KLF2 regulon (***Figure 6A and B***, ***Figure 1—figure supplement 3***, ***Figure 6—figure supplement 1***). This demonstrated that iNKT cell subset diversity is preserved after antigen challenge. Among resting iNKT cells, however, two new clusters emerged, expressing *Maf* or *Klrg1* (***Figure 6A and B***), matching our reactivated data. Independent of our clustering, we also detected mutually exclusive gene-level expression of *Maf* and *Klrg1* among resting iNKT cells (***Figure 6C***), consistent with two distinct iNKT cell populations expressing *Maf* or *Klrg1*. We confirmed this finding using flow cytometry, identifying significant enrichment of mutually exclusive cMAF⁺ and KLRG1⁺ iNKT cell populations among resting versus steady-state splenic iNKT cells (***Figure 6D and E***). This demonstrates that cMAF⁺ and KLRG1⁺ iNKT cells are induced after antigen challenge. Interestingly, we found that the frequency of cMAF⁺ iNKT cells was positively correlated with antigen load (***Figure 6—figure supplement 2***), suggesting that TCR signal strength may regulate cMAF expression in iNKT cells, a signaling axis which has previously been reported in γδ T cells (***Zuberbuehler et al., 2019***).

We next performed gene expression analysis to analyze the transcriptional profile of cMAF⁺ and KLRG1⁺ iNKT cells. cMAF⁺ cells displayed enrichment of memory and exhaustion markers associated

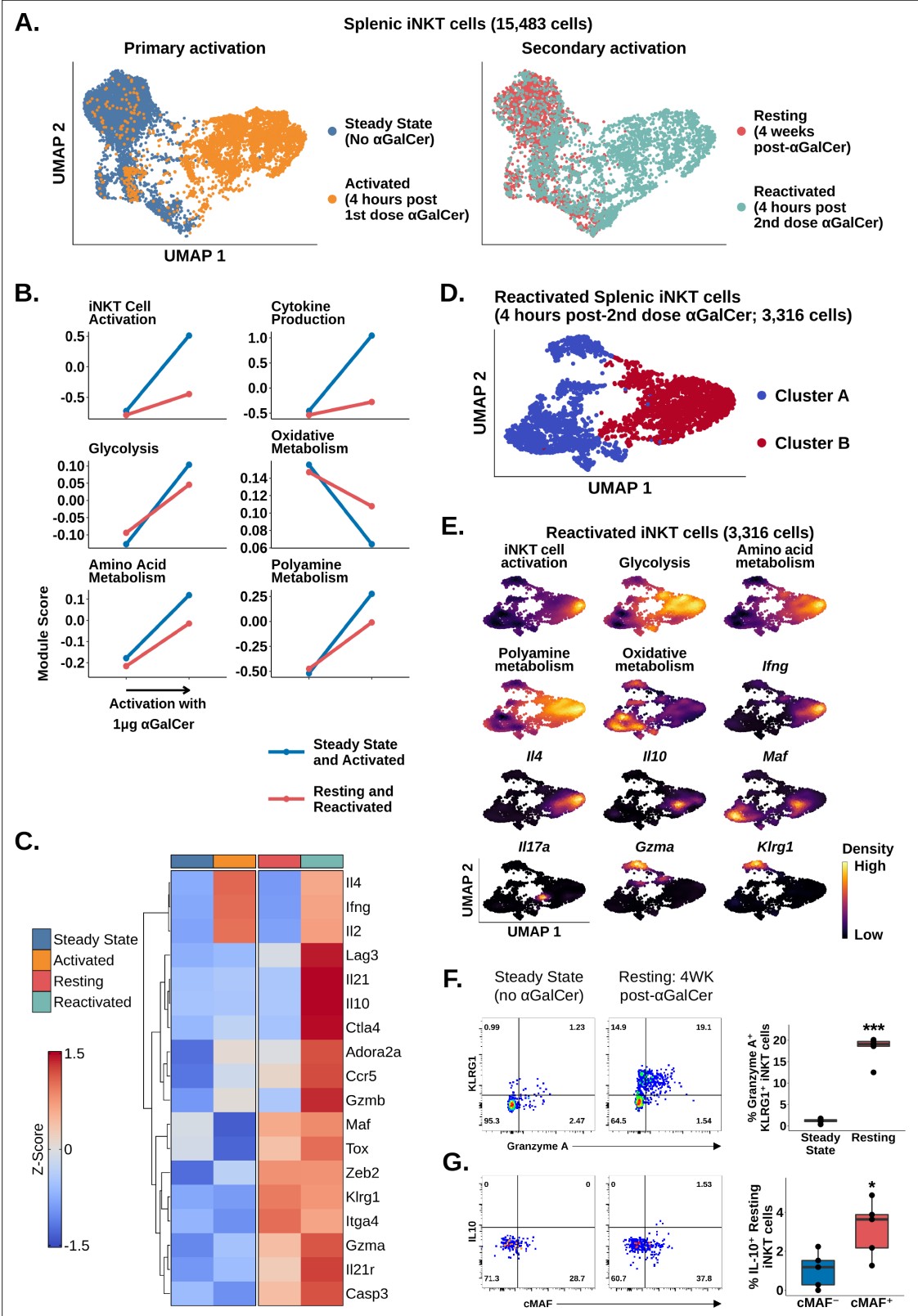

**Figure 5.** Chronic activation of splenic invariant natural killer T (iNKT) cells induces an adipose-like phenotype and the appearance of populations expressing Tr1 cell markers. (**A**) Uniform manifold approximation and projection (UMAP) of murine splenic iNKT cells. (**B**) Line plots showing median expression of functional and metabolic gene module scores in the data from (**A**). (**C**) Heatmap of scaled averaged gene expression with hierarchical clustering in the data from (**A**). (**D**) UMAP of subclustered murine splenic reactivated iNKT cells from (**A**). (**E**) Density plots of functional and metabolic

*Figure 5 continued on next page*

*Figure 5 continued*

gene module score and gene expression in the reactivated iNKT cells from the data in (**D**). (**F**) Representative pseudocolor plots of Granzyme A versus KLRG1 expression in steady state and resting splenic iNKT cells at 4 weeks post-α-galalctosylceramide (post-αGalCer) and after restimulation with 50 ng phorbol 12-myristate 13-acetate (PMA) and 1 μg ionomycin for 4 hr ex vivo (left), and box plot quantification of the pseudocolor plot data (right). iNKT cells were defined as live, single CD45$^+$ CD8$^-$ CD19$^-$ F4/80$^-$ CD3$^{low}$ CD1d-PBS57 tetramer$^+$ cells. N = 5 biological replicates from one experiment. Experiment performed at least twice. Student's unpaired *t*-test. Asterisks denote significance, *p<0.05; **p<0.01; ***p<0.001. Source data provided in *Figure 5—source data 1*. The central box plot horizontal line denotes the median value. (**G**) Representative pseudocolor plots of cMAF versus IL-10 expression in steady state and resting splenic iNKT cells at 4 weeks post-αGalCer and after restimulation with 50 ng PMA and 1 μg Ionomycin for 4 hr ex vivo (left), and box plot quantification of the resting iNKT cell data from the pseudocolor plot data (right). iNKT cells were defined as live, single CD45$^+$ CD8$^-$ CD19$^-$ F4/80$^-$ CD3$^{low}$ CD1d-PBS57 tetramer$^+$ cells. N = 5 biological replicates from one experiment. Experiment performed at least twice. Student's unpaired *t*-test. Asterisks denote significance, *p<0.05; **p<0.01; ***p<0.001. Source data provided in *Figure 5—source data 2*. The central box plot horizontal line denotes the median value.

The online version of this article includes the following source data and figure supplement(s) for figure 5:

**Source data 1.** % IL-10 production by flow cytometry in cmaf postive or negative iNKT cells.

**Source data 2.** % KLRG1 Granzyme A expression by flow cytometry by iNKT without or without treatment.

**Figure supplement 1.** Reanalysis from GSE47959 showing gene expression differences between splenic iNKT with and outout a previous aGalCer activation.

**Figure supplement 2.** Activation markers in iNKT cells at different activation states.

**Figure supplement 3.** Functional and metabolic gene module scores in different iNKT cell clusters.

**Figure supplement 4.** Distinct expression patterns of MAF, IL10, Granzyme and KLRG1 associated with different iNKT cell states.

with CD4$^+$ memory T cells, stem-like memory T cells, Tregs, and T precursor exhausted cell (TPEX) populations, including *Cd4, Slamf6, Tcf7, Ctla4, Lag3, Cxcr3,* and *Tox* (*Figure 6B*; *Andreatta et al., 2021*; *Utzschneider et al., 2016*; *Jadhav et al., 2019*). We also detected enrichment of *Zbtb16*, *Izumo1r* (FR4), and the Tr1 cell markers *Il27ra, Maf,* and *Hif1a* (*Figure 6B*; *Chihara et al., 2016*; *Mascanfroni et al., 2015*; *Apetoh et al., 2010*). By contrast, KLRG1$^+$ cells showed enrichment of cytotoxic T cell, effector memory CD8$^+$ T cell, and NK cell markers such as *Gzmb, Gzma, Klrg1, Ncr1, Klrd1, S1pr5,* and *Klre1* (*Figure 6B*). KLRG1$^+$ cells also expressed *Spry2* and *Cx3cr1* (*Figure 6B*), which were previously identified in KLRG1$^+$ iNKT cells by *Murray et al., 2021*, and the transcription factor *Zeb2*. ZEB2 is a key transcription factor regulating terminal differentiation of KLRG1$^+$ CD8$^+$ effector cells (*Omilusik et al., 2015*), suggesting that ZEB2 may be a candidate regulator of KLGR1$^+$ iNKT cells. cMAF$^+$ and KLRG1$^+$ cells both demonstrated enrichment of the antigen experience marker *Itga4* (*Shimizu et al., 2014*; *Grau et al., 2018*), suggesting that these two populations are memory-like or 'trained'. Interestingly, whole-transcriptome correlation analysis revealed that cMAF$^+$ cells and KLRG1$^+$ cells were more similar to NKT1 cells than to NKT2 or NKT17 cells (*Figure 6—figure supplement 1*), and we noted that production of Granzyme A and IL-10 by iNKT cells at 4 weeks post-αGalCer, which we previously associated with KLRG1$^+$ and cMAF$^+$ iNKT cells, respectively (*Figure 5*), was also associated with co-production of IFNγ (*Figure 6—figure supplement 3*). Conversely, we did not detect co-production of IL-17A and IL-10 (*Figure 6—figure supplement 4*). This suggests that cMAF$^+$ and KLRG1$^+$ cells are NKT1-like populations and/or that cMAF$^+$ and KLRG1$^+$ cells may differentiate from NKT1 cells.

Having identified memory-like splenic cMAF$^+$ and KLRG1$^+$ iNKT cell populations that appear after αGalCer challenge, we wondered whether analogous populations also appeared in other organs after αGalCer challenge. Using flow cytometry, we identified a significant enrichment of KLRG1$^+$ iNKT cells at 4 weeks post-αGalCer in the lung, liver, adipose tissue, and inguinal lymph nodes (*Figure 6D and E*), while cMAF$^+$ iNKT cells were enriched in the liver, adipose tissue, and inguinal lymph nodes (*Figure 6D and E*). scRNA-seq also revealed distinct cMAF$^+$ and KLRG1$^+$ iNKT cell populations among 3014 adipose iNKT cells at 4 weeks post-αGalCer (*Figure 6—figure supplement 5*), indicating conserved enrichment of cMAF$^+$ and KLRG1$^+$ iNKT cells after αGalCer challenge across different tissues. Interestingly, we noted that minor populations of cMAF$^+$ and KLRG1$^+$ iNKT cells were present in the adipose tissue at steady state (*Figure 6D and E*), which correlated with our previous finding that adipose iNKT cells constitutively express markers of antigen experience and chronic activation (*Figure 3*). We have previously shown that adipose NK1.1$^-$ iNKT cells express *Klrg1, Maf,* and *Itga4* at steady state (*LaMarche et al., 2020*), and subclustering of steady-state adipose NK1.1$^-$ iNKT cells revealed distinct cMAF$^+$-like or KLRG1$^+$-like iNKT cell subpopulations at the transcriptional level

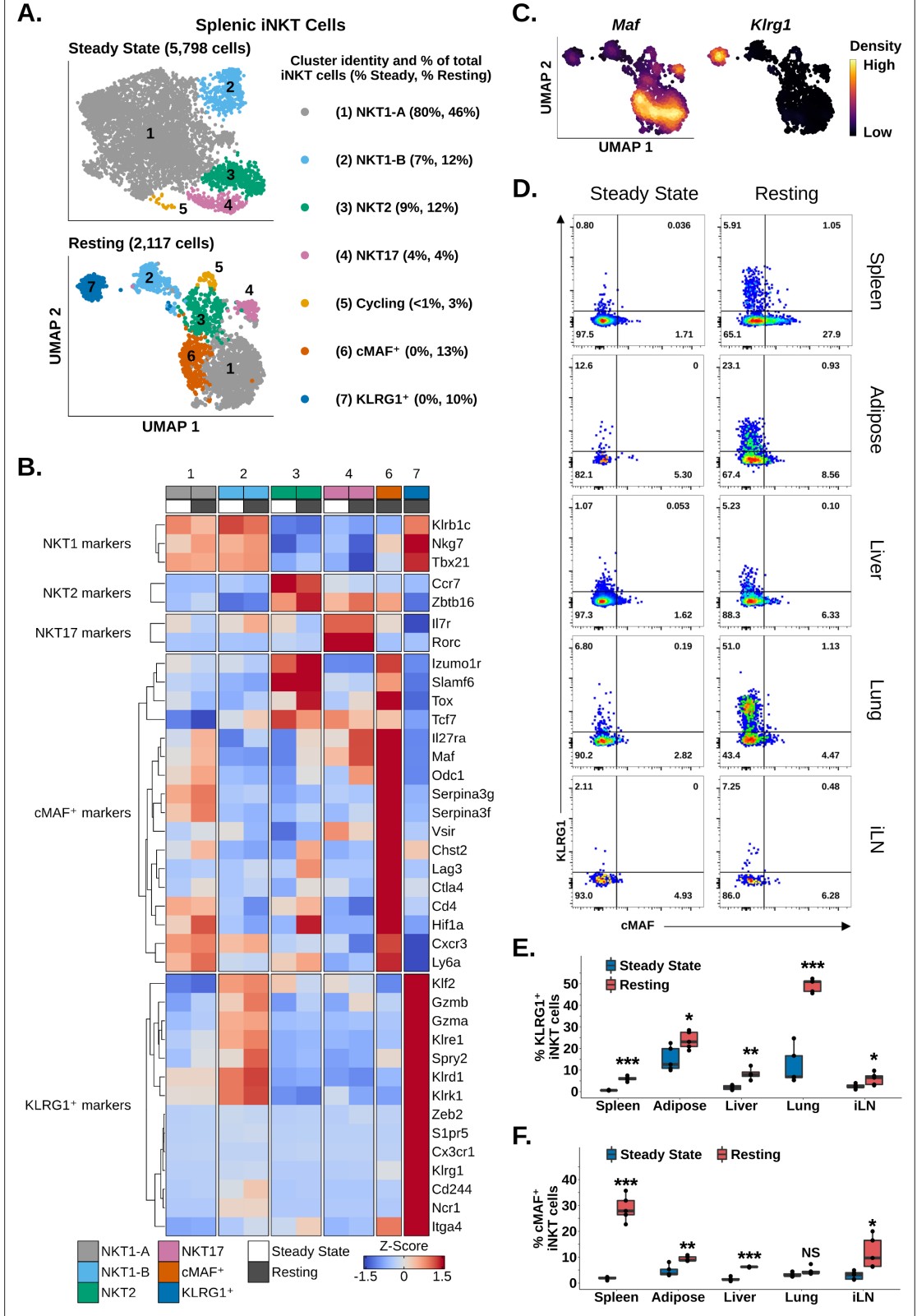

**Figure 6.** Memory-like cMAF[+] and KLRG1[+] invariant natural killer T (iNKT) cells are induced in the spleen following α-galalctosylceramide (αGalCer) challenge, and similar populations are constitutively present in adipose tissue. (**A**) Uniform manifold approximation and projection (UMAP) of murine splenic iNKT cells. (**B**) Heatmap of scaled averaged gene expression with hierarchical clustering in the data from (**A**). Both datasets were merged and normalized together, and cycling cells were excluded from the analysis. (**C**) Density plots showing mutually expression of *Maf* and *Klrg1* in the data from

*Figure 6 continued on next page*

*Figure 6 continued*

(**A**) (bottom UMAP). (**D**) Representative pseudocolor plots of RORγT (spleen, adipose tissue, lung, and inguinal lymph nodes) or total (liver) murine splenic iNKT cells at steady-state or resting iNKT cells at 4 weeks post-αGalCer. iNKT cells were defined as live, single CD19- CD8- F4/80- CD3low CD1d-PBS57 tetramer+ cells. iLN, inguinal lymph node. (**E**) Box plots quantifying the KLRG1 percentage data from (**D**). N = 4 biological replicates for hepatic iNKT cells at 4 weeks post-αGalCer and N = 5 biological replicates for all other data from one experiment. Experiment performed at least twice in the spleen and once for other tissues. Student's unpaired *t*-test. Asterisks denote significance, * Padj<0.05; ** Padj<0.01; *** Padj<0.001. Source data provided in *Figure 6—source data 1*. The central box plot horizontal line denotes the median value. (**F**) Box plots quantifying the cMAF percentage data from (**D**). N = 4 biological replicates for hepatic iNKT cells at 4 weeks post-αGalCer and N = 5 biological replicates for all other data from one experiment. Experiment performed at least twice in the spleen and once for other tissues. Student's unpaired *t*-test. Asterisks denote significance, * Padj<0.05; ** Padj<0.01; *** Padj<0.001. Source data provided in *Figure 6—source data 1*. The central box plot horizontal line denotes the median value.

The online version of this article includes the following source data and figure supplement(s) for figure 6:

**Source data 1.** Gene expression patterns in resting iNKT, 4 weeks post stimulation.

**Figure supplement 1.** Gene expression of spenic iNKT cells resting 4 weeks post activation.

**Figure supplement 2.** Strength of TCR signal is associated with strength of MAF expression.

**Figure supplement 3.** Comparison of splenic and adipose iNKT cells expression of functional markers.

**Figure supplement 4.** IL17 and IL10 expression is mutually exclusive on iNKT cells.

**Figure supplement 5.** Resting adipose iNKT cells 4 weeks post aGalCer activation.

---

(*Figure 6—figure supplement 5*). Overall, these data indicate that cMAF+ and KLRG1+ iNKT cell populations are induced across multiple organs in response to antigenic stimulation and are constitutively present in adipose tissue.

## Identification of a conserved cMAF-associated and NKT$_{FH}$-like transcriptional state in NKT10 cells

Having identified transcriptional signatures of regulatory iNKT cells in adipose tissue and after serial antigen activation, we next sought to describe shared transcriptional features of these different NKT10 cell populations. Gene expression analysis identified 110 genes enriched among splenic NKT10 cells (*Supplementary file 7*), including *Ctla4*, *Pdcd1*, *Lag3*, *Il21*, *Maf*, *Hif1a*, and *Ccr5* (*Figure 7A*), all of which were already identified in adipose NKT10 cells. We identified 39 genes conserved across NKT10 cells from both tissues (*Figure 7B*), including Tr1 cell markers, *Tgfb1*, and the tolerogenic factors *Slfn2* and *Vsir* (*Berger et al., 2010*; *ElTanbouly et al., 2020*; *Figure 7B*). We also found that splenic NKT10 cells expressed the adipose iNKT cell marker *Nfil3* (*Figure 7A*), which we previously linked to IL-10 production by regulatory adipose NKT10 cells (*Lynch et al., 2015*; *LaMarche et al., 2020*; *Motomura et al., 2011*). However, gene regulatory network analysis using GENIE3 (*Huynh-Thu et al., 2010*) of *Il10* versus *Nfil3* and other transcription factors identified in NKT10 cells revealed that *Maf* demonstrated the greatest correlation with *Il10* in NKT10 cells (*Figure 7C*), matching our previous transcriptional and functional analysis correlating IL-10 with cMAF (*Figure 5*). Since cMAF is known to regulate IL-10 in other immune populations, such as Tr1 cells, B cells, and macrophages[57,66,67], our data suggest that *Maf* is a major candidate regulator of NKT10 cells. Our analysis also indicates that cMAF+ iNKT cells are a memory-like population of NKT10 cells or that NKT10 cells are significantly enriched among cMAF+ iNKT cells.

We next sought to compare the transcriptional signature of NKT10/cMAF+ cells against other memory-like iNKT cell populations. Interestingly, module scoring of our resting scRNA-seq data demonstrated that NKT10/cMAF+ cells but not KLRG1+ cells showed enrichment for NKT$_{FH}$ cell gene signatures (*Figure 7D*). Reanalysis of published bulk iNKT cell RNA-seq data 6 days post-αGalCer (GSE161492) (*Murray et al., 2021*) also revealed that NKT$_{FH}$ cells but not steady-state or non-NKT$_{FH}$ cells expressed NKT10/cMAF+ markers, including *Maf*, *Cd4*, *Tox*, *Izumo1r*, *Slamf6*, *Hif1a*, and *Lag3* (*Figure 7E*, *Supplementary file 7*), and we found co-enrichment of NKT10/cMAF+ and NKT$_{FH}$ markers in published microarray data comparing αGalCer-pretreated versus steady-state splenic iNKT cells (GSE47959) (*Figure 5—figure supplement 1*, *Figure 7—figure supplement 1*, *Supplementary file 8*). These data indicated that NKT10/cMAF+ cells are transcriptionally similar to NKT$_{FH}$ cells, and suggested that these two memory-like iNKT cell populations might overlap. However, we found that most cMAF+ iNKT cells did not co-express the NKT$_{FH}$ markers CXCR5 and PD-1 (*Figure 7F and G*), and

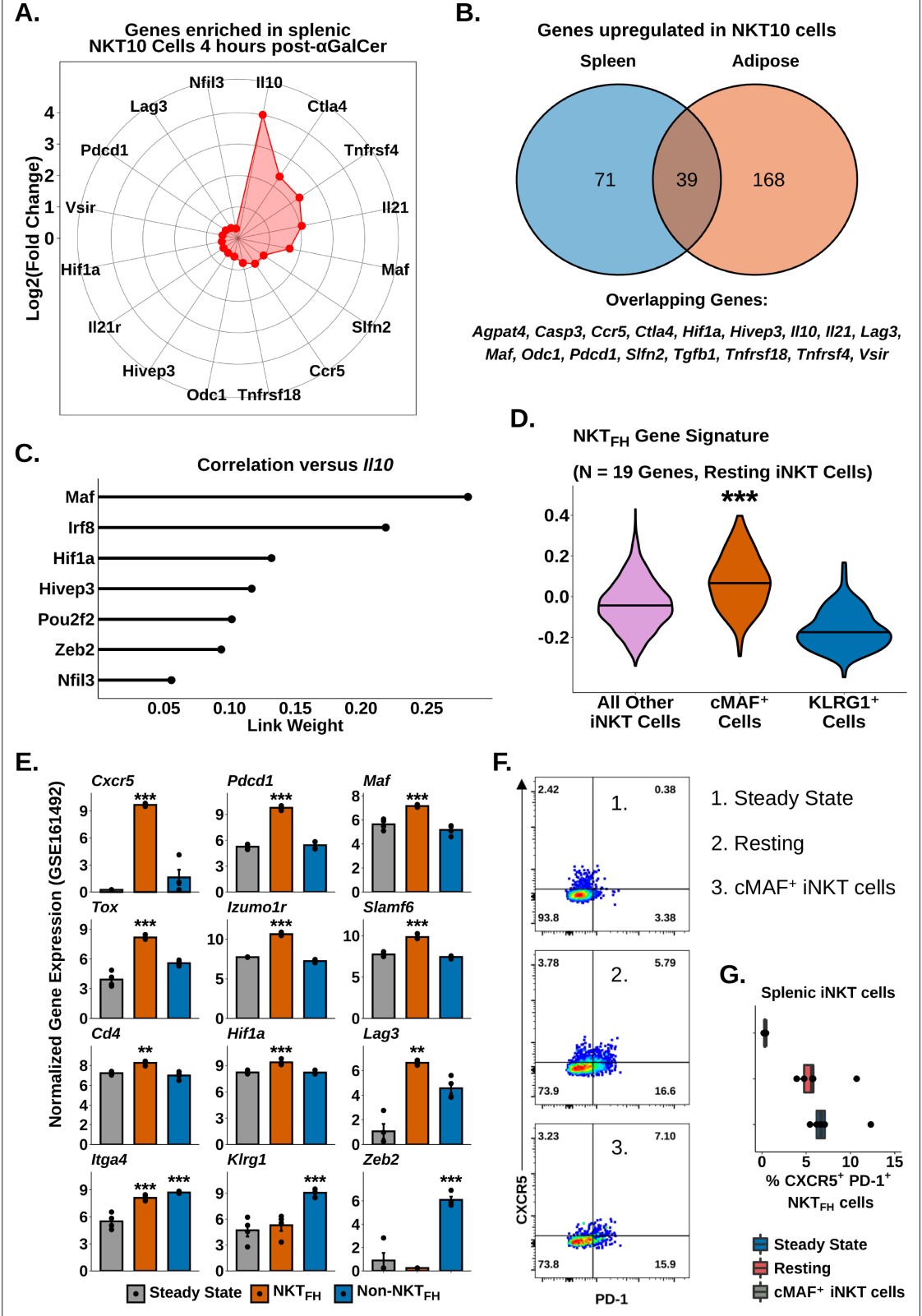

**Figure 7.** Identification of a conserved cMAF-associated and NKT$_{FH}$-like transcriptional state in NKT10 cells. (**A**) Radar chart showing Log2(Fold Change) values of genes enriched in murine splenic activated $Il10^{pos}$ invariant natural killer T (iNKT) cells versus activated $Il10^{neg}$ iNKT cells at 4 hr post-α-galalctosylceramide (post-αGalCer). (**B**) Venn diagram showing overlap of genes enriched in murine splenic (left) and adipose (right) $Il10^{pos}$ iNKT cells after αGalCer (activated 4 hr post-αGalCer splenic iNKT cells, and 4 hr and 72 hr post-αGalCer for adipose iNKT cells). (**C**) Lollipop plot showing

*Figure 7 continued on next page*

*Figure 7 continued*

correlation (Link Weight) values of different transcription factors versus *Il10* from GENIE3 analysis of total splenic and adipose NKT10 cells. Source data provided in *Figure 7—source data 1*. (D) Violin plots showing expression and enrichment of NKT$_{FH}$ signature gene module scoring in resting splenic iNKT cells, with NKT1-A, NKT1-B, NKT2, NKT17 and cycling cell clusters pooled together (all other iNKT cells) and compared versus cMAF$^+$ and KLRG1$^+$ iNKT cell clusters. The central violin plot horizontal line denotes the median value. Asterisks denote significance, * Padj<0.05; ** Padj<0.01; *** Padj<0.001. (E) Bar plots showing bulk RNA-seq gene expression values in untreated total murine splenic iNKT cells (steady state) and murine splenic NKT$_{FH}$ or non-NKT$_{FH}$ cells (6 days post-αGalCer). Reanalyzed from GSE161492. Asterisks indicate significantly increased expression versus all other populations or versus steady state alone (*Itga4* only). * Padj<0.05; ** Padj<0.01; *** Padj<0.001. (F) Representative pseudocolor plots of CXCR5 versus PD-1 expression in murine total steady-state splenic iNKT cells (1), total resting splenic iNKT cells at 4 weeks post-αGalCer (2) or splenic RORγT cMAF$^+$ iNKT cells at 4 weeks post-αGalCer (3). iNKT cells were defined as live, single CD19$^-$ CD8$^-$ F4/80$^-$ CD3$^{low}$ CD1d-PBS57 tetramer$^+$ cells. (G) Box plots quantifying the data from (F). N = 5 biological replicates from one experiment. Experiment performed once. Source data provided in *Figure 7—source data 2*. The central box plot horizontal line denotes the median value.

The online version of this article includes the following source data and figure supplement(s) for figure 7:

**Source data 1.** Gene expression in iNKT cell clusters.

**Source data 2.** NKT follicular helper cell associated gene expression in iNKT cell subsets.

**Figure supplement 1.** NKTFH signature gene expression in control and αGalCer-pretreated splenic iNKT cells at rest.

we only detected a minor population of CXCR5$^+$ PD-1$^+$ NKT$_{FH}$ cells among cMAF$^+$ iNKT cells (~7% of cells, *Figure 7F and G*). We also did not detect enrichment of the flagship NKT$_{FH}$ transcription factor BCL6 in cMAF$^+$ iNKT cells either by flow cytometry or by scRNA-seq (*Figure 7—figure supplement 1*). Therefore, our data indicate that while NKT10/cMAF$^+$ iNKT cells are transcriptionally similar to NKT$_{FH}$ cells, most NKT10/cMAF$^+$ iNKT cells are not bone fide NKT$_{FH}$ cells, and NKT10/cMAF$^+$ cells represent a distinct lineage of memory-like iNKT cells induced following activation with αGalCer.

## Discussion

iNKT cells and other innate T cells, such as γδ T cells and MAIT cells, rapidly activate following antigen or cytokine stimulus and produce potent cytokine responses. In this study, we characterized the transcriptional programs underpinning the iNKT cell response to antigen, revealing an initial phase of rapid cytokine production, *Zbtb16* upregulation, and metabolic gene remodeling, which was followed by second phase of proliferation, further remodeling of metabolic gene programs, and acquisition of features associated with memory-like iNKT cells. This transcriptional framework was highly conserved across iNKT cells from different tissues and species, and between functionally distinct iNKT cell subsets, including NKT1, NKT2, and NKT17 cells. Interestingly, we identified many similarities with adaptive T cell activation, including early induction of biosynthesis and aerobic glycolysis gene signatures (*Hartmann et al., 2021*), although in keeping with the poised phenotype of iNKT cells this occurred over hours instead of days, in concert with innate-like cytokine production, and expression of metabolic activation and cytokine production gene signatures was largely diminished by 3 days post-activation.

We sequenced 48,813 iNKT cells, the largest number of iNKT cells analyzed by scRNA-seq to date, which allowed us to explore iNKT cell heterogeneity in unprecedented detail. We found that NKT2 and NKT17 cells, despite sharing many features of activation with NKT1 cells, were more enriched for genes associated with oxidative metabolism after activation, and production of NKT2 and NKT17 cell cytokines was more dependent on oxidative phosphorylation compared to NKT1 cell cytokines. Recent work by *Raynor et al., 2020* has shown that NKT2 and NKT17 cells are more enriched for oxidative metabolism than NKT1 cells during thymic development, and downregulation of oxidative metabolism was required for establishment of an NKT1 cell phenotype (*Raynor et al., 2020*). Here, we show that that enrichment of oxidative metabolism persists in splenic NKT2 and NKT17 cells after thymic development and defines their function, and we also identified enrichment of oxidative gene signatures in adipose NKT17 cells. Interestingly, we have shown that γδ17 cells display enrichment of oxidative metabolic signatures (*Lopes et al., 2021*), suggesting that enrichment of oxidative metabolism may be a conserved feature of IL-17-producing innate T cells.

Our study found that regulatory iNKT cell populations exhibited a blunted and/or delayed response to αGalCer, coupled with increased expression of early exhaustion and regulatory Tr1 cell markers. This agrees with previous studies demonstrating long-term anergy in iNKT cells treated with αGalCer (*Parekh et al., 2005*; *Sag et al., 2014*) and enrichment of regulatory iNKT cells after prior αGalCer

challenge (*Sag et al., 2014*). Interestingly, we found that adipose iNKT cells, which are constitutively enriched for NKT10 cells, showed evidence of chronic endogenous activation, and repeated αGalCer challenge induced an adipose-like phenotype in splenic iNKT cells. Thus, chronic activation can promote a type of iNKT cell anergy, characterized by enrichment of regulatory iNKT cells. Chronic activation is known to promote exhaustion and anergy in adaptive T cells (*Liu et al., 2019*; *Seo et al., 2019*), and a recent study by *Vorkas et al., 2022* found that chronically activated human MAIT cells upregulate expression of FOXP3 and early exhaustion markers such as PD-1 and LAG-3 (*Vorkas et al., 2022*). Functionally, induction of regulatory and/or anergic innate T cells following chronic activation could serve as a protective mechanism to prevent overactivation of innate T cells and limit immunopathogenicity. This could be relevant in autoimmunity, and it is notable that chronic activation of iNKT cells with αGalCer has been shown to ameliorate experimental autoimmune encephalomyelitis (*Jahng et al., 2001*; *Singh et al., 2001*).

Characterization of regulatory iNKT cells revealed two distinct memory-like iNKT cell populations, cMAF+ cells and KLRG1+ cells, which were enriched across multiple organs after prior antigen exposure and constitutively present in adipose tissue regardless of activation status. KLRG1+ iNKT cells showed enrichment of gene signatures associated with cytotoxic effector memory CD8+ T cells and NK cells, and expressed the transcription factor *Zeb2*. By contrast, cMAF+ iNKT cells were transcriptionally similar to CD4+ memory T cells, stem-like memory T cells, precursor exhausted T cells, Tregs and NKT$_{FH}$ cells, and were enriched for IL-10-producing NKT10 cells. Although cMAF+ and NKT$_{FH}$ cells were similar at the transcriptional level, we found that most cMAF+ cells could not be classified as NKT$_{FH}$ cells, indicating that cMAF+ iNKT cells represent a distinct lineage of memory-like iNKT cells. However, given the transcriptional similarity between cMAF+ and NKT$_{FH}$ cells, it is possible these populations may follow a similar differentiation trajectory. Moreover, given the similarity between KLRG1+ cells and CD8+ effector/memory T cells, and CD4+ memory T cells, cMAF+ cells and NKT$_{FH}$ cells, we hypothesize that there could be a bifurcating differentiation branch during the formation of memory-like iNKT cell populations, whereby a CD8+ effector/memory-like KLRG1+ iNKT cell population differentiates along one trajectory, and CD4+ memory-like cMAF+ and/or NKT$_{FH}$ cell populations differentiate along another trajectory. Mechanistically, it is possible that some of the same transcription factors that control the differentiation of CD8+ vs. CD4+ memory could regulate the development of different memory-like iNKT cell lineages. For example, *Zeb2* is known to be a key regulator of the terminal differentiation of effector memory T CD8+ cells (*Omilusik et al., 2015*; *Evrard et al., 2022*) and could play a similar role in KLRG1+iNKT cell differentiation. Further research will be required to elucidate the dynamics of memory-like iNKT cell differentiation in more detail.

We have shown that E4BP4 (*Nfil3*) rather than FOXP3 regulates production of IL-10 by adipose iNKT cells (*Lynch et al., 2015*; *LaMarche et al., 2020*), and we identified *Nfil3* expression among adipose and splenic NKT10 cells. However, most adipose iNKT cells express E4BP4 and do not express IL-10, suggesting that other factors must regulate IL-10. Furthermore, E4BP4 has been shown to bind an intronic region of the IL-10 promoter (*Motomura et al., 2011*), suggesting that other promoter-associated transcription factors are likely required to effectively induce IL-10 production in iNKT cells. Here, we implicate the AP-1 family transcription factor cMAF as a candidate regulator of NKT10 cells, correlating with published literature identifying a role for cMAF in regulation of IL-10 production by macrophages, B cells, and adaptive T cells, including Tr1 cells (*Chihara et al., 2016*; *Apetoh et al., 2010*; *Liu et al., 2018*; *Cao et al., 2005*). cMAF is also expressed by NKT17 cells and is required for their development (*Thapa et al., 2017*), as well as for γδ17 cell development (*Zuberbuehler et al., 2019*). However, although NKT17 cells can produce IL-10 after in vitro expansion (*Cameron and Godfrey, 2018*), we and others did not identify co-expression of IL-10 and IL-17 among in vivo or ex vivo iNKT cells (*Sag et al., 2014*). Interestingly, RORγT has been demonstrated to suppress production of IL-10 in Th17 cells (*Sun et al., 2019*), suggesting that it could perform a similar role in iNKT cells. We found that NKT10 cells co-expressed *Ifng*, *Il4*, and *Il21* with *Il10*, and lacked *Rorc* expression, indicating that NKT10 cells are more similar to NKT1 cells and NKT$_{FH}$ cells than NKT17 cells. Notably, Murray et al. showed that most NKT$_{FH}$ cells arise from T-bet+ NKT1 cells; given the transcriptional similarity of NKT10 and NKT$_{FH}$ cells, it is possible NKT10 cells also arise from an NKT1 cell lineage. We also noted that NKT10 cells were transcriptionally similar to Tr1 cells, which are regulatory but lack expression of FOXP3, and are known to express cMAF (*Yan et al., 2017*; *Gruarin et al., 2019*; *Chihara et al., 2016*; *Alfen et al., 2018*; *Mascanfroni et al., 2015*; *Apetoh et al., 2010*). Therefore, our data

suggest that NKT10 cells may be similar to Tr1 cells (*Yan et al., 2017*; *Chihara et al., 2016*; *Alfen et al., 2018*). Tr1 cells are heterogeneous, and we found that expanded adipose NKT10 cells were heterogeneous, with some cells expressing *Il4* and *Il21*, and other cells expressing *Gzmb*.

Understanding the factors controlling the phenotype and generation of regulatory iNKT cells has a several potential applications. For example, NKT10 cells have been shown to emerge in tumors, where they can promote tumor growth and regulatory T cell function (*Wang et al., 2018b*). However, our work indicates that NKT10 cells are capable of co-expressing IFNγ and IL-10. Skewing regulatory iNKT cells toward a less regulatory phenotype by enhancing IFNγ and inhibiting IL-10 might have a therapeutic benefit. We have shown that regulatory adipose iNKT cells can induce weight loss and suppress adipose tissue inflammation (*Lynch et al., 2016*), and are enriched in human adipose tissue (*Lynch et al., 2009*), suggesting that modulation of adipose iNKT cells might be beneficial during obesity and metabolic syndrome. Insights into the factors and transcriptional signatures governing iNKT cell activation are also potentially relevant to other innate T cell populations, as γδ T cells and MAIT cells respond similarly after activation (*Godfrey et al., 2015*; *Vorkas et al., 2020*), and the transcriptomic resource that we present here may assist other studies investigating these innate T cell responses.

## Methods

### In vivo stimulations

αGalCer (Avanti Polar Lipids) was prepared by dissolving in sterile DMSO (Sigma) at a concentration of 1 mg/mL. αGalCer was prepared for injection by dilution in sterile phosphate-buffered saline (PBS) (Sigma), and the final concentration of DMSO was adjusted to less than 10% vol/vol. Mice were injected IP with 1 µg of αGalCer or PBS vehicle. Mice were sacrificed and tissues harvested after either 4 or 72 hr. For analysis of memory-like iNKT cells, mice were injected IP with 4 µg of αGalCer and sacrificed after 4 weeks, with some animals receiving an additional IP injection of 1 µg of αGalCer 4 hr before sacrifice. All injections had a final volume of 100 µL.

### Tissue processing

Adipose tissue was excised, minced with a razor, and digested in 1 mg/mL Collagenase Type II (Worthington) in RPMI shaking for 25–30 min at 37°C. Digested cells were filtered through a 70 µM nylon mesh and centrifuged at 15,000 rpm for 5–7 min to pellet the stromovascular fraction (SVF). Spleens and thymi were disrupted through a 70 µM filter and pelleted. Red blood cells in the spleen and thymus were lysed with ACK Lysing Buffer (VWR) or RBC Lysis Buffer (BioLegend) prior to further analysis.

### Ex vivo stimulations and mitochondrial staining

Where indicated, thymocytes, splenocytes, and the isolated SVF from adipose tissue were cultured for 4 hr in the presence of PMA and ionomycin (Cell Stimulation Cocktail, BioLegend) and Brefeldin A (BioLegend), and in the presence or absence of oligomycin (Sigma) or Monesin (BioLegend). Cultures were in complete RPMI media supplemented with L-glutamine, penicillin, streptomycin, and 10% FBS (Thermo Fisher Scientific). For mitochondrial staining, thymocytes and splenocytes were cultured for 30 min in complete RPMI media containing TMRM (Invitrogen) and/or MitoTracker Green FM (Thermo Fisher Scientific).

### Flow cytometry and cell sorting

All antibody staining of live cells was performed in PBS (Gibco) with 1–2% FBS. Single-cell suspensions were incubated in Fc-receptor blocking antibody (Clone 93, BioLegend) during cell surface antigen staining. Dead cells were excluded with Fixable Viability dyes (UV, eFluor 780; Thermo Fisher Scientific) or Zombie Aqua (BioLegend). For intracellular antigen staining, cells were fixed with either True-Nuclear Transcription Factor Buffer Set (BioLegend) for 45 min at room temperature, Foxp3/ Transcription Factor Fixation/Permeabilization kit (Thermo Fisher Scientific), or Cytofix/Cytoperm kit (BD Biosciences) for 30 min at room temperature. The following anti-mouse antibodies were obtained from BioLegend: anti-CD3 (17A2), anti-CD19 (6D5), anti-CD11b (M1/70), anti-CD45 (30-F11), anti-TCRβ (H57-597), anti-CD8a (53–6.7), anti-KLRG1 (2F1/KLRG1), anti-IL-17A (TC11-18H10.1), anti-IL-4

(11B11), anti-IFNγ (XMG1.2), anti-CD43 Activation Glycoform (1B11), anti-ICOS (C398.4A), and anti-CXCR5 (L138D7). The following anti-mouse antibodies were obtained from Thermo Fisher Scientific: anti-cMAF (sym0F1), anti-RORγT (B2D), anti-RORγT (AFKJS-9), anti-CD8a (53–6.7), anti-IL-10 (JES5-16E3), anti-IL-13 (eBio13A), anti-CD11b (M1/70), anti-F4/80 (BM8), anti-BCL6 (BCL-DWN), and anti-Granzyme A (GzA-3G8.5). The following anti-mouse antibodies were obtained from BD Biosciences: anti-IFNγ (XMG1.2). iNKT cells were identified as live, single lymphocytes binding to anti-TCRβ or anti-CD3 antibodies and αGalCer analog PBS57-loaded CD1d tetramer (NIH Tetramer Core Facility/Emory Vaccine Center). A 'dump' channel with antibodies against CD19, or CD19 and CD11b or F4/80, or CD19, CD8a, and CD11b or F4/80 was used to eliminate nonspecific staining. For staining of BALB/c thymic iNKT cell subsets, NKT1 cells were gated as CD43-HG$^-$ ICOS$^-$ CD3$^{low}$ cells, NKT2 cells were gated as CD43-HG$^-$ ICOS$^+$ CD3$^{high}$ cells, and NKT17 cells were gated as CD43-HG$^+$ cells. Samples were acquired using LSR Fortessa and FACS Canto II cytometers and sorted using a FACS Aria Fusion Cell Sorter. Flow cytometry analysis and plots were created using FlowJo version 10.0.7r2.

## scRNA-seq sequencing and data preprocessing

scRNA-seq was performed on single-cell suspensions of sorted iNKT cells from the visceral adipose tissue and spleens of mice using the 10X Genomics platform. A total of 35 visceral adipose tissue deposits from 35 mice or 5 spleens from 5 mice were pooled for each scRNA-seq sample. Nine biological samples were sequenced in three batches (*Supplementary file 9*). For two of the batches, adipose and/or splenic iNKT cell samples were first tagged by TotalSeq-A Mouse hashtag antibodies (BioLegend) and then pooled for sequencing. Cell suspensions were loaded onto a 10x Chromium Controller to generate single-cell Gel Beads-in-emulsion (GEMS) and GEMs were processed to generate UMI-based libraries according to the 10X Genomics Chromium Single Cell 3' protocol. Libraries were sequenced using a NextSeq 500 sequencer (Illumina). Raw BCL files were demultiplexed using Cell Ranger v3.0.2 mkfastq to generate fastq files with default parameter. Fastq files were aligned to the mm10 genome (v1.2.0) and feature reads were quantified simultaneously using Cell Ranger count for feature barcoding. The resulting filtered feature-barcode UMI count matrices containing quantification of gene expression and hashtag antibody binding were then utilized for downstream analysis.

## Downstream scRNA-seq data analysis

A total of 48,813 cells murine iNKT cells expressing a minimum median of 1567 genes per cell and 4163 UMIs per cell were loaded from feature-barcode UMI count matrices using the Seurat v4.0.3 package (*Hafemeister and Satija, 2019*). scRNA-seq data for steady-state adipose iNKT cells and splenic iNKT cells at steady state and 4 hr post-αGalCer were previously uploaded to GSE142845 (*LaMarche et al., 2020*). Raw human iNKT cell scRNA-seq data was downloaded from GSE128243 (*Zhou et al., 2020*). Antibody hashtag data was demultiplexed using the Seurat HTODemux function with a positive quantile of 0.99 and centered log ratio transformation normalization. Cells positive for more than one antibody were removed from the analysis. Genes expressed in less than three cells were excluded from all analyses to prevent false-positive identification of transcripts. Cells expressing less than 1% minimum or more than 12.5% maximum of mitochondrial genes as a % of total gene counts were considered to represent empty droplets or apoptotic/dead cells and were removed from the analysis. Cells were also filtered based on total UMI counts and total gene counts on a per-sample basis to remove empty droplets, poor quality cells and doublets, with a minimum cutoff of at least 500 genes per cell across all samples. UMI counts were normalized using regularized negative binomial regression using sctransform v0.3.2 (*Hafemeister and Satija, 2019*). Where indicated, cycle regression was performed by first normalizing UMI counts using sctransform, then performing cell cycle scoring using the CellCycleScoring() function and cell cycle gene lists provided with the Seurat package, and then re-normalizing raw RNA count data with sctransform and regression of computed cell cycle scores applied.

Dimensionality reduction was performed using principal component analysis (PCA) with n = 100 dimensions and 2000 or 3000 variable features, and an elbow plot was used to determine the number of PCA dimensions used as input for UMAP (*McInnes et al., 2018*). For collective analysis of cells from different batches, the harmony v1.0 package (*Korsunsky et al., 2019*) was used with default settings to remove batch effects, and batch-corrected harmony embeddings were used for UMAP. Outlier cells

expressing genes associated with macrophages (e.g., *Adgre1*, *Cd14*), B cells (e.g.,. *Cd19*), or CD8+ T cells (e.g., *Cd8a*) were also identified, and these cells were removed prior to the final analysis. UMAP was performed using a minimum distance of 0.3 and a spreading factor of 1. Shared nearest neighbor (SNN) graphs were calculated using k = 20 nearest neighbors. Graph-based clustering was performed using the Louvain algorithm. In some cases, overclustering was performed and clusters were manually collapsed, and/or the first two dimensions of the UMAP reduction were used as input for graph-based clustering instead of PCA or harmony embeddings. Steady-state splenic NKT1, NKT2, and NKT17 cell identification was performed using expression of *Tbx21*, *Zbtb16*, and *Rorc* after graph-based clustering. Cycling cells were identified using expression of *Mki67*. Activated NKT1, NKT2, and NKT17 cell identification was performed using expression of *Ifng*, *Il4*, *Il13*, *Il17a*, and *Il17f* after graph-based clustering. Identified NKT1, NKT2, and NKT17 cells from steady state and 4 hr post-αGalCer datasets were combined by subset, and low-level graph-based clustering was performed to stratify 'true' steady state and activated iNKT cells into separate steady state or activated clusters. Reclustered steady state and activated iNKT cells from different subsets were then recombined and renormalized for final downstream analysis.

Gene expression analysis was performed using the FindMarkers() or FindAllMarkers() Seurat functions and the Wilcoxon rank-sum test. All gene expression analyses were performed using log-normalized RNA counts. Gene set enrichment analysis (GSEA) (*Subramanian et al., 2005*) was performed using FGSEA v1.17.0 (*Korotkevich et al., 2016*) and clusterProfiler v3.99.2 (*Yu et al., 2012*) packages. KEGG (*Kanehisa et al., 2021*) pathway data was retrieved from the Molecular Signatures Database (MSigDB) (*Liberzon et al., 2011*) using the msigdbr v7.4.1 package. Over-representation analysis was performed using g:Profiler (*Raudvere et al., 2019*). For cross-dataset differential gene expression analysis of adipose versus splenic iNKT cells, individual gene expression analyses were first performed between adipose and splenic iNKT cells at steady state, 4 hr post-αGalCer and 72 hr post-αGalCer. Genes significantly enriched in either adipose or splenic iNKT cells across all three analyses were identified, which included 971 genes enriched among adipose iNKT cells and 65 genes enriched among splenic iNKT cells. Over-representation analysis was then performed on these enriched genes using gprofiler. Heatmaps were generated using the Complex Heatmap v2.7.13 and circlize v0.4.13 packages (*Gu et al., 2016*). Module scores were calculated using the AddModuleScore() Seurat function with n = 10 control features. Density plots were produced using the Nebulosa v1.1.1 package (*Alquicira-Hernandez and Powell, 2021*). Gene regulatory network analysis was performed using GENIE3 v1.14.0 (*Huynh-Thu et al., 2010*) with n = 10 iterations and link weight values were averaged between replicate analyses. Other plots were created using egg v0.4.5, GGally v2.1.2, ggiraphExtra v0.3.0, ggpubr v0.4.0, pals v1.7, patchwork v1.1.1, tidyverse v1.3.1, tidymodels v0.1.3, and viridis v0.6.1.

## Bulk RNA-seq and microarray analysis

Raw RNA-seq count files were downloaded from GEO Repository GSE161492 (*Murray et al., 2021*). Microarray data was downloaded from GEO Repository GSE47959 (*Sag et al., 2014*). Raw CEL files were annotated against the Mouse430_2 Array (mouse4302.db) and Robust Multichip Average (RMA) normalized using the affy v1.7.0 R package. Discrete probes corresponded to the same gene were merged and values were averaged. Raw RNA-seq counts were transformed using the cpm() function with log = TRUE and trimmed mean of M-values (TMM) normalized with edgeR using edgeR v3.33.7 (*Robinson et al., 2010*). Genes with low read counts were filtered out using the edgeR filterByExpr() function. Testing for differential gene expression was performed with Limma-Voom using limma v3.47.16 (*Smyth, 2005*; *Law et al., 2014*) with standard settings and without a minimum fold change cutoff. Plots were generated using the same libraries as used for scRNA-seq data plotting.

## Statistics

Sample size for adequate power was determined based on previous studies (*Lynch et al., 2015*; *LaMarche et al., 2020*). Significance was determined by Student's two-tailed *t*-test with Holm–Bonferroni correction, or one-way ANOVA with Tukey's post hoc, where indicated. Significance is presented as *p<0.05, **p<0.01, ***p<0.001, with p>0.05 considered nonsignificant. All statistical analyses were performed using rstatix v0.7.0, stats v4.1.2, and ggpubr v0.4.0 R packages or GraphPad Prism v9.4.1.

All sequencing data analyses were performed using R 4.1.2 and RStudio Desktop v1.4.1712 on an Ubuntu 20.04 Linux GNU (64 bit) system.

## Acknowledgements

The authors thank the Brigham and Women's Hospital Single Cell Genomics Core for assistance with sequencing and preprocessing of scRNA-seq data, the NIH Tetramer Core Facility for recombinant CD1d-PBS57 tetramers, and A.T. Chicoine for cell sorting. This work was supported by NIH grants R01 AI134861, American Diabetes Association 1-16-JDF-061 and ERC Starting grant 14283 (Project ID: 679173) (to LL), and R01 AI113046 (to MBB). Cartoons were created with BioRender.com.

## Additional information

### Funding

| Funder | Grant reference number | Author |
| --- | --- | --- |
| American Diabetes Association | 1-16-JDF-061 | Lydia Lynch |
| National Institutes of Health | R01AI134861 | Lydia Lynch |
| European Research Council | 679173 | Lydia Lynch |
| Science Foundation Ireland | | Harry Kane Lydia Lynch |
| National Institutes of Health | AI113046 | Michael B Brenner |

The funders had no role in study design, data collection and interpretation, or the decision to submit the work for publication.

### Author contributions

Harry Kane, Conceptualization, Data curation, Formal analysis, Validation, Investigation, Visualization, Methodology, Writing - original draft, Writing – review and editing; Nelson M LaMarche, Formal analysis, Validation, Investigation, Writing – review and editing; Áine Ní Scannail, Validation, Investigation; Amanda E Garza, Data curation, Project administration; Hui-Fern Koay, Data curation, Formal analysis, Methodology; Adiba I Azad, Brenneth Stevens, Data curation, Formal analysis; Britta Kunkemoeller, Data curation; Michael B Brenner, Resources, Funding acquisition, Writing – review and editing; Lydia Lynch, Conceptualization, Resources, Software, Supervision, Funding acquisition, Methodology, Project administration, Writing – review and editing

### Author ORCIDs

Lydia Lynch http://orcid.org/0000-0002-4273-4681

### Ethics

Male C57BL/6 mice were purchased from Jackson Laboratory or Harlan Laboratory. BALB/c mice were purchased from Harlan Laboratory. Male and female BALB/c mice were bred under specific-pathogen-free facilities at Trinity College Dublin. Mice were used in experiments at 6–14 weeks of age. All animal work was approved and conducted in compliance with the Trinity College Dublin University Ethics Committee and the Health Products Regulatory Authority Ireland, and the Institutional Animal Care and Use Committee guidelines of The Dana Farber Cancer Institute and Harvard Medical School. C57BL/6 mice were used for all experiments unless otherwise specified. Balb/c mice were used for surface staining of markers delineating NKT subsets so that co-staining for mitochondrial markers was possible.

### Decision letter and Author response

Decision letter https://doi.org/10.7554/eLife.76586.sa1
Author response https://doi.org/10.7554/eLife.76586.sa2

## Additional files

### Supplementary files

• Supplementary file 1. Gene expression data associated with the scRNA-seq data in *Figure 1B*.

• Supplementary file 2. Lists of genes used for module scoring of scRNA-seq data, starting with *Figure 1D*.

• Supplementary file 3. Gene expression data associated with the scRNA-seq data in *Figure 2A*.

• Supplementary file 4. Lists of conserved gene expressed in adipose or splenic invariant natural killer T (iNKT) cells, associated with *Figure 3B*.

• Supplementary file 5. Lists of conserved gene expressed in adipose NKT10 cells, associated with *Figure 4E*.

• Supplementary file 6. Lists of conserved gene expressed in adipose NKT10 cells, associated with *Figure 7A*.

• Supplementary file 7. Gene expression data associated with reanalysis of bulk RNA-seq data from GSE161492 in *Figure 7E*.

• Supplementary file 8. Gene expression data associated with reanalysis of bulk microarray data from GSE47959, first referenced in *Figure 5—figure supplement 1*.

• Supplementary file 9. Information about scRNA-seq data batch processing and sample information, associated with the 'Methods' section.

• Transparent reporting form

### Data availability

Sequencing data have been deposited in GEO under accession code GSE190201.

The following dataset was generated:

| Author(s) | Year | Dataset title | Dataset URL | Database and Identifier |
|---|---|---|---|---|
| Kane H, LaMarche NM, Brenner MB, Lynch L | 2021 | Single cell analysis of activated iNKT cells from murine epididymal adipose tissue and spleen | https://www.ncbi.nlm.nih.gov/geo/query/acc.cgi?acc=GSE190201 | NCBI Gene Expression Omnibus, GSE190201 |

The following previously published datasets were used:

| Author(s) | Year | Dataset title | Dataset URL | Database and Identifier |
|---|---|---|---|---|
| LaMarche NM, Kane H, Kohlgruber AC, Lynch L, Brenner MB | 2020 | Single cell analysis of iNKT cells from murine epididymal adipose tissue and spleen | https://www.ncbi.nlm.nih.gov/geo/query/acc.cgi?acc=GSE142845 | NCBI Gene Expression Omnibus, GSE142845 |
| Wang J, Adrianto I, Wu X, Zhou L, Mi Q | 2020 | Single-cell RNA-seq of human peripheral blood NKT cells | https://www-ncbi-nlm-nih-gov.ezproxy.u-pec.fr/geo/query/acc.cgi?acc=GSE128243 | NCBI Gene Expression Omnibus, GSE128243 |
| Wingender G, Kronenberg M | 2014 | NKT-10 cells represent a novel invariant NKT cell subset with regulatory characteristics | https://www.ncbi.nlm.nih.gov/geo/query/acc.cgi?acc=GSE47959 | NCBI Gene Expression Omnibus, GSE47959 |

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
