## [Editor Report]

This study presents a valuable finding on transcriptional profiles of various subsets of activated invariant natural killer T (iNKT) cells using longitudinal scRNA-seq analysis. The evidence supporting the conclusions is solid with rigorous and thorough bioinformatic analyses. The work will be of interest to scientists within the field of iNKT cells.

---

## [Decision Letter]

**Decision letter after peer review:**

Thank you for submitting your article "Longitudinal analysis of invariant natural killer T cell activation reveals a cMAF-associated transcriptional state of NKT10 cells" for consideration by *eLife*. Your article has been reviewed by 3 peer reviewers, one of whom is a member of our Board of Reviewing Editors, and the evaluation has been overseen by Tadatsugu Taniguchi as the Senior Editor. The reviewers have opted to remain anonymous.

*Reviewer #1 (Recommendations for the authors):*

1. Based on scRNA-seq data, the authors should address whether TCR repertoire of iNKT cells is altered at different stages of activation (e.g., in the steady, activation, and re-activation states).

2. In Figure 5D, the authors identified two major populations of reactivated splenic iNKT cells (cluster A and cluster B). What is the interrelationship of these two clusters of iNKT cells? Does the proportion of KLRG1+ and c-MAF+ iNKT cells change at various time points after ⍺-GalCer immunization (e.g., 3 days, 7 days, 30 days, and reactivation)? Does activation with ⍺-GalCer-pulsed DC also induce these two unique iNKT cell populations?

3. In Figure 6C, it is worthwhile to determine the frequency of KLRG1+ and c-MAF+ iNKT cells in other tissues (e.g., liver, lymph nodes) at steady and resting state by flow cytometry.

4. The authors should compare the function of activated and reactivated iNKT cells in the adipose tissues.

5.The authors suggested that "NKT10/cMAF+ cells are transcriptionally similar to NKTFH cells, and these two memory-like iNKT cell populations may phenotypically and functionally overlap". Does NKT10/cMAF+ cells also express CXCR5, PD-1 and Bcl6?

6. The authors should discuss how ⍺-GalCer immunization leads to the emergence of two distinct lineages of memory-like iNKT cells. Does TCR signaling strength affect the c-MAF expression in iNKT cells?

*Reviewer #2 (Recommendations for the authors):*

1. Please add reference from the Bendelac lab re: parabiosis and tissue residence. J Exp Med. 2011 Jun 6;208(6):1179-88.

2. In Figure 1C, the authors noted that some activated iNKT cells express genes of the KLF2 regulon and suggest that activated iNKT cells may traffic to other tissues. This should be contrasted to the work of the Kubes and Littman labs, which showed that activated iNKT cells stop patrolling blood vessels.

3. In Figure 1D, the analysis of metabolic pathway is interesting. However, I would consider toning down some of the conclusions at the data is limited to gene expression and pathway analyses. Functional metabolic studies are necessary to draw firm conclusions.

4. The experimental setting in Figure 2H should be detailed. Were subsets sorted prior to in vitro stimulation, or were total thymic cells stimulated? Were iNKT subsets discriminated by FACS using transcription factor staining on top of cytokine antibodies. A similar experiment should be done with cells from the spleen. This experiment used Balb/c mice. Was the in vivo activation performed earlier in the manuscript performed with Balb/c or C57Bl/6 mice? The background of mice used should be clarified for all experiments. In B6 mice, iNKT1 cells are known to produce IL-4 in addition to IFN-g. Is IL-4 production by iNKT1 cells also more affected by oligomycin than their production of IFN-g?

5. The authors state: "We previously found that Nur77 is enriched in adipose iNKT cells compared to splenic iNKT cells at steady state (17), but Nur77 expression does not increase upon activation in adipose iNKT cells, unlike splenic iNKT cells. Is this based on the transcriptomic data, or has this been confirmed by FACS using Nur77 antibodies, or Nur77 reporter mice?

6. The blunted initial response observed for adipose tissue iNKT cells could rather be indicative of a different kinetic of activation between spleen and adipose tissue, due to sub-anatomical location of iNKT cells in these tissues, and accessibility to aGalCer and CD1d (see work from the Hogquist lab). A fine kinetic study of adipose tissue vs. spleen iNKT cell response may be warranted to draw this conclusion. This is in line with the statement drawn from Figure 3D-E that adipose tissue iNKT cells are more activated at baseline. Maybe they respond faster following aGalCer administration than their spleen counterparts.

7. The suggestion (from supplementary Figure 3) that iNKT10 might be functionally heterogenous is interesting and the data seems convincing. However, this could be due to subtle differences in the activation kinetic in vivo. Is there a way to fate-map some of these genes following aGalCer activation in vivo?

8. On Figure 5A, it seems to me that steady-state and resting (4 weeks post-aGalCer) iNKTs cluster together while activated and reactivated iNKTs cluster together. The "greatly reduced response" and the similarity with activated adipose tissue iNKTs is no obvious from this UMAP data. The authors should clarify this.

9. In Figure 6A, please clearly describe in the text what markers were used to define iNKT subsets among steady-state and resting I NKT cells.

10. How do the clusters A-C described in Figure 6G compare with effector subsets described in Figure 4?

11. "Flow cytometry of splenic iNKT cells 4 weeks post-⍺GalCer confirmed that IL-10+ iNKT cells expressed cMAF, and cMAFneg cells produced little IL-10 (Figure 7F, Figure 7G)." These panels do not appear to show FACS data. Does this relate to panel D? It would be better to show IL-10 vs. cMAF from CD1d tetramer-positive cells on these plots rather than CD1d tetramer staining. The figure should also use PMA/Iono stimulation of steady-state iNKT cells from the same tissue as a control.

*Reviewer #3 (Recommendations for the authors):*

This paper is of interest to scientists within the field of iNKT cells. The authors conducted scRNA-seq to longitudinally profile activated iNKT cells and generated a transcriptomic atlas of iNKT cells at the activation states. The study suggests that transcriptional signatures of activation are highly conserved among heterogeneous iNKT cell populations and that the adipose iNKT cells undergo blunted activation and display constitutive enrichment of memory like population. In addition, the study also identifies a conserved cMAF- associated network in NKT10 cells. The data quality is relatively high.

1. Based on the violin plots shown in Figure 2B, in the 4 hours aGalCer stimulation group, it seems that Tbx21 and Zbtb16 are evenly spaced in all three clusters, where it is difficult to identify NKT1, NKT2 in this case. It would be clearer if the author could show NKT_1/2_/17 signatures expression pattern in umap plot.

2. The statement " Notably, we found that all subsets upregulated Zbtb16 and Tbx21 after activation (Figure 2B), suggesting that PLZF and T-bet play subset-independent roles during activation." This needs more evidence. The gene expression level (such as Tbx21, Zbtb16) increased post aGalCer activation could be due to the transcriptional bursting, author should show and compare the house keeping genes expression level in both steady state and aGalCer activation state.

3. In Figure 2D, using correlation heatmaps the authors mapped the gene profiles in different subsets of NKT cells before and after aGalCer stimulation, and claimed that "activated NKT2 and NKT17 cells are transcriptionally similar compared to NKT1 cells." The authors should consider that post aGalCer stimulation, proliferating genes/cells will be predominant. The similarity of NKT2 and NKT17 observed in the heatmap could be due to the overwhelming cell cycle genes, instead of the intrinsic genes of NKT2 and NKT17. The authors should remove cell cycle gene, and then do the mapping.

4. The study claimed that NKT10 and Tr1 share transcriptional features. It will make this study more interesting if the author could employ the experiments to check on their function potential.

5. Figure 2F only shows the representation of the experiment, statistics summary bar graphs should also be included.

6. Figure 3B is generated from bulk analysis of scRNA-Seq data, right? What about by Stages (4 and 72h) subclusters?

7. "Analysis of cytokine production among adipose iNKT cells revealed that IL-10 was only expressed by NKT1 cells (Figure 4B/D)." Better have flow data to support this claim. No scale stick in Y axis in Figure 4C.

8. Clusters showing in Figure 5F with NKT 10 cells are better validated in protein level by FACS.

9. Clusters showing in Figure 6G should be validated in protein level.

10. The author did not mention how many biological replicates for scRNA-seq experiment, since mice treated with activator could vary widely between mouse to mouse, at least 2 biological replicates are needed.

11. The author mentioned in the method that both C57BL6 mice and BALB/c mice were used in this study, but did not mention which experiments were performed using C57BL6 mice or BALB/c. NKT cell subsets are different between C57BL6 and BALB/C mice. Please discuss the rationale why these two strains were used and make them clearer.

---

## [Author Response]

Reviewer #1 (Recommendations for the authors):1. Based on scRNA-seq data, the authors should address whether TCR repertoire of iNKT cells is altered at different stages of activation (e.g., in the steady, activation, and re-activation states).

The idea of checking if and how the TCR repertoire of iNKT cells changes during activation is an excellent one, and we thank the reviewer for this recommendation. However, from a bioinformatic perspective, robust TCR repertoire analysis using scRNA-Seq typically requires that TCR sequencing be performed in combination with standard scRNA-Seq at the time of data generation. Since TCR repertoire analysis was not a specific aim of this study we did not perform TCR-Seq at the time of sequencing. Also, given the restricted nature of the iNKT cell TCR, we didn’t feel that it was a key factor when designing the experiment initially. Therefore, we respectfully suggest that scRNA-Seq-based analysis of iNKT cell TCR repertoires is not within the scope of this study.

2. In Figure 5D, the authors identified two major populations of reactivated splenic iNKT cells (cluster A and cluster B). What is the interrelationship of these two clusters of iNKT cells? Does the proportion of KLRG1+ and c-MAF+ iNKT cells change at various time points after ⍺-GalCer immunization (e.g., 3 days, 7 days, 30 days, and reactivation)? Does activation with ⍺-GalCer-pulsed DC also induce these two unique iNKT cell populations?

We thank the reviewer for these questions, which we also found to be interesting. To answer these questions, we performed new experiments by flow cytometry.

In Figure 5D of our original manuscript we identified two main clusters or populations of reactivated splenic cells, Cluster A and Cluster B. The main difference between these two clusters mainly was their differential expression of activation markers, cytokine genes, and transcripts associated with metabolic remodeling and T cell metabolic activation (Figure 5D5E, Figure 5—figure supplement 3). Cluster A iNKT cells (~50% of reactivated iNKT cells) demonstrated low expression of activation-associated genes compared to Cluster B iNKT cells (~50% of reactivated iNKT cells; Figure 5D-5E, Figure 5—figure supplement 3). The expression of activation-associated genes in Cluster A iNKT cells was comparable to that of resting iNKT cells (gray histogram) which were not reactivated with αGalCer (Figure 5figure supplement 3). We also performed subclustering analysis of the iNKT cells in Cluster A and Cluster B. Within Cluster B we identified distinct populations of iNKT cells expressing genes associated with NKT1 cells (*IFNγ* and *Il4*), NKT17 cells (*Il17a*), cMAF^+^ iNKT cells (*Maf*, *Il10*) and KLRG1^+^ iNKT cells (*Klrg1*, *Gzma*) (Figure 5E, Figure 5—figure supplement 3). This indicated that Cluster B is comprised of a heterogeneous mix of different iNKT cell populations. Interestingly, we also identified distinct iNKT cell populations expressing *Klrg1* or *Maf* within Cluster A (Figure 5E, Figure 5—figure supplement 3), which suggested to us that prior activation of iNKT cells with αGalCer might induce the differentiation or development of memory-like or “trained” KLRG1^+^ and cMAF^+^ iNKT cell populations. In the absence of any evidence to suggest that Cluster A iNKT cells are significantly different to Cluster B iNKT cells in any respect other than in their expression of activation markers and cytokines, we hypothesize that the cells in Cluster A would eventually assume a more activated transcriptional profile, similar to the cells in Cluster B.

To improve our data presentation, and to comply with reviewer recommendations, we have reworked Figure 5 in our revised manuscript. We have replaced the previously separate density plots of gene expression in Cluster A and Cluster B cells (Figure 5F and 5G in the original manuscript) with density plots of gene expression and module scores for all reactivated iNKT cells in a single UMAP (Figure 5E). We believe that this updated plot provides a more direct and improved visualization of the data. The histograms and density plots from Figure 5E, 5F and 5G of the original manuscript are now presented in Figure 5figure supplement 3. We also present new flow cytometry data (Figure 5F, Figure 5G, Figure 5—figure supplement 4) to support our gene expression analysis, focusing on our identification of co-expression of *Klrg1* and *Gzma*, and *Maf* and *Il10* (Figure 5E, Figure 5—figure supplement 4). In our revised Figure 5F we identify a significant increase in the frequency of Granzyme A^+^ KLRG1^+^ iNKT cells among reactivated versus activated splenic iNKT cells (Figure 5F). Similarly, we show that there is significantly increased production of IL-10 in cMAF^+^ versus cMAF^-^ resting splenic iNKT cells (Figure 5G) and significantly increased expression of cMAF among resting splenic iNKT cells (Figure 5—figure supplement 4).

The frequency of KLRG1^+^ iNKT cells after αGalCer stimulation has previously been assayed by Shimzu *et al.* (2014) and Murray *et al.* (2021), demonstrating that KLRG1 expression peaks between 5-7 days post-αGalCer and thereafter declines, although both our study and previous studies have demonstrated that KLRG1^+^ iNKT cells can persist for weeks to months after αGalCer stimulation^1,2^. Shimizu *et al.* (2014) identified KLRG1^+^ iNKT cells after immunization of mice with αGalCer-pulsed dendritic cells, indicating that memory-like or “trained” iNKT cell populations can be induced using this method of immunization. In our study we identified increased expression of *Klrg1* among iNKT cells at 72 hours post-αGalCer compared to 4 hours post-αGalCer or steady state (Figure 1, Figure 1—figure supplement 3), consistent with previous data indicating that the frequency of KLRG1^+^ iNKT cells peaks several days post-αGalCer. We also found that expression of cMAF was significantly increased among iNKT cells at 72 hours post-αGalCer (Figure 1, Figure 1—figure supplement 3), suggesting that cMAF^+^ iNKT cells may also be induced at this time point. To validate this observation, we analyzed expression of cMAF among iNKT cells at 72 hours post-αGalCer using flow cytometry. Interestingly, we found that ~50% of iNKT cells expressed cMAF (Figure 1—figure supplement 3), a larger proportion of cMAF^+^ iNKT cells than we had identified at 4 weeks post-αGalCer (Figure 6D and Figure 6E). Interestingly, we also noted that this high percentage of cMAF^+^ iNKT cells at 72 hour post-αGalCer correlated with increased production of IL-10 after restimulation of iNKT cells at 72 hours post-αGalCer compared to primary stimulation of steady state iNKT cells (Figure 1—figure supplement 3). Therefore, our study demonstrates, in conjunction with published data, that the frequency of cMAF^+^ and KLRG1^+^ iNKT cells changes at different time points following αGalCer immunization, peaking several days post αGalCer, but remaining albeit at lower levels in some iNKT cells at 4 weeks post αGalCer.

3. In Figure 6C, it is worthwhile to determine the frequency of KLRG1+ and c-MAF+ iNKT cells in other tissues (e.g., liver, lymph nodes) at steady and resting state by flow cytometry.

We agree and have performed a new experiment to examine the frequency of KLRG1^+^ and cMAF^+^ iNKT cells at steady state and 4 weeks after αGalCer in different tissues using flow cytometry. We found that KLRG1^+^ and cMAF^+^ iNKT cells are present at low levels at steady state and enriched 4 weeks after αGalCer (Figure 6D, Figure 6E), in all organs tested including spleen, lung, liver, adipose tissue, and inguinal lymph nodes. KLRG1 and cMAF expression was mutually exclusive across the organs (Figure 6D-6F). Interestingly, we found that KLRG1^+^ iNKT cells, but not cMAF^+^ iNKT cells, were specifically enriched in the lung 4 weeks post-αGalCer (Figure 6D-6E), correlating with previous work by Shimzu *et al.* (2014) which demonstrated that KLRG1^+^ iNKT cells are strongly induced in the lung after immunization of mice with αGalCer-pulsed dendritic cells.

4. The authors should compare the function of activated and reactivated iNKT cells in the adipose tissues.

We thank the reviewer for this recommendation. We have previously profiled the function of activated adipose iNKT cells in detail (Figure 2 of LaMarche *et al.* (2020), DOI: https://doi.org/10.1016/j.cmet.2020.05.017)^3^ and here we add to that with new data on the function of reactivated adipose iNKT cells in response to the reviewer’s question. Restimulated adipose iNKT cells at 4 weeks post-αGalCer demonstrated mutually exclusive production of Granzyme A and IL-10 (Figure 5—figure supplement 4), and expression of Granzyme A was mainly associated with KLRG1^+^ iNKT cells (Figure 5—figure supplement 4). We also identified a similar expression pattern in splenic iNKT cells (Figure 5E). Granzyme A has previously been shown to be a flagship secreted factor produced by KLRG1^+^ iNKT cells^1^ and here we demonstrate that production of IL-10 is enriched among cMAF^+^ iNKT cells at 4 weeks post-αGalCer (Figure 5F). Therefore, our functional analysis of Granzyme A and IL-10 production supports our KLRG1 and cMAF expression data in the adipose tissue and the spleen (Figure 6D-6F).

Interestingly, we found that most Granzyme A^+^ iNKT cells and IL-10^+^ iNKT cells produced IFNγ (Figure 6—figure supplement 3). Similarly, we found that IL-17A and IL-10 expression was mutually exclusive in the spleen at 4 weeks post-αGalCer. We did not detect robust production of IL-17A in the adipose tissue at 4 weeks post-αGalCer, which matched our scRNA-Seq data demonstrating that there are very few RORyT^+^ NKT17 cells in the adipose tissue at 4 weeks post-αGalCer (Figure 6—figure supplement 5). This matches previous functional data on induced NKT10 cells published by Sag *et al.* (2014)^4^ and suggests that KLRG1^+^ iNKT cells and iNKT10 cells are NKT1-like populations (or were previously NKT1 cells). Surprisingly, we did not detect robust production of IL-21 among adipose or splenic iNKT cells at 4 weeks post-αGalCer, even though *Il21* was one of the most differentially expressed genes among reactivated splenic iNKT cells and NKT10 cells by scRNA-Seq and in published microarray data from Sag *et al.* (2014)^4^ (Figure 5C, Figure 7A-7B, Figure 4—figure supplement 1, Figure 5—figure supplement 1, Figure 7—figure supplement 1). This could suggest that IL-21 is only expressed at the protein level under different activation circumstances. Further research will be required to elucidate if and when IL-21 might be expressed by reactivated iNKT cells or NKT10 cells.

5. The authors suggested that "NKT10/cMAF+ cells are transcriptionally similar to NKTFH cells, and these two memory-like iNKT cell populations may phenotypically and functionally overlap". Does NKT10/cMAF+ cells also express CXCR5, PD-1 and Bcl6?

We thank the reviewer for this important insight which we have now examined. Investigating the relationship between NKT_FH_ cells and cMAF^+^ iNKT cells, we found a minority of splenic iNKT cells co-expressing the NKT_FH_ cell markers CXCR5 and PD-1 at 4 weeks post-αGalCer (Figure 7F, Figure 7G; Resting; ~6% of iNKT cells on average). The average percentage of CXCR5^+^ PD-1^+^ NKT_FH_ cells was much lower than the percentage of cMAF^+^ iNKT cells in the same gate (Figure 6D, Figure 6F, ~28% of iNKT cells on average), which indicated that most cMAF^+^ iNKT cells were unlikely to be NKT_FH_ cells based on expression of CXCR5 and PD-1. Gating on cMAF^+^ iNKT cells, a minority (~7%) coexpressed CXCR5 and PD-1 (Figure 7F, Figure 7G), although this was a slight increase relative to total resting iNKT cells (Figure 7G, a ~1.2 fold increase in the frequency of NKT_FH_ cells among cMAF^+^ iNKT cells versus total resting iNKT cells). We did not find a significant increase in BCL6 expression among cMAF^+^ iNKT cells versus total resting iNKT cells (Figure 7—figure supplement 1), matching our scRNA-Seq where *Bcl6* was not significantly enriched among cMAF^+^ iNKT cells (Figure 7—figure supplement 1). This suggests that there is some overlap between NKT_FH_ cells and cMAF^+^ iNKT cells, and that NKT_FH_ cells are slightly enriched among cMAF^+^ iNKT cells, but most cMAF^+^ iNKT cells are not *bone fide* NKT_FH_ cells. At the gene level, however, cMAF^+^ iNKT cells express many of the same genes as NKT_FH_ cells (Figure 7D, Figure 7E), suggesting that these two populations are transcriptionally similar. This result and discussion are presented in the revised manuscript.

6. The authors should discuss how ⍺-GalCer immunization leads to the emergence of two distinct lineages of memory-like iNKT cells. Does TCR signaling strength affect the c-MAF expression in iNKT cells?

We thank the reviewer for their recommendation. We have added to the discussion to address how αGalCer leads to the emergence of two distinct lineages of memory-like iNKT cells. In brief, we note that KLRG1^+^ iNKT cells display enrichment of markers associated with NK cells, cytotoxic T cells and effector memory CD8^+^ T cells, including *Klrg1*, *Gzma*, *Klf2*, *Gzmb*, and *Zeb2*^5–7^ (Figure 6B). This similarity of KLRG1^+^ iNKT cells to cytotoxic T cells and effector memory CD8^+^ T cells has previously been noted by Shimizu *et al.* (2014)^1^, and a similar comparison was also made between KLRG1-like nonNKT_FH_ cells (termed NKT_eff_ cells) and effector memory CD8^+^ T cells/cytotoxic T cells by Murray *et al.* (2021)^2^. We also note that the transcriptional signature/phenotype of KLRG1^+^ iNKT cells is similar to that of short lived effector CD8^+^ T cells (SLECs) associated with the SLEC/memory precursor effector cell (MPEC) model described in numerous studies^6–8^. By contrast, we found that cMAF^+^ iNKT cells and NKT10 cells display a gene signature more similar to CD4^+^ memory T cells, Tregs, precursor exhausted T (TPEX) cells and MPECs, including *Maf*, *Cd4*, *Il10*, *Tox*, *Cxcr3*, *Slamf6* and *Izumo1r^7–10^* (Figure 6B, Figure 7A, Figure 7B). Therefore, we propose that there may be a bifurcating differentiation branch during the formation of memory-like iNKT cell subsets following immunization with αGalCer, whereby a CD8^+^ effector memory-like KLRG1^+^ iNKT cell population differentiates along one trajectory and a CD4^+^ memory-like T cell cMAF^+^ iNKT cell population differentiates along another trajectory. Furthermore, given the transcriptional similarity of NKT_FH_ cells and cMAF+ iNTK cells, it is possible that these two populations might share a similar differentiation trajectory. Mechanistically, it is possible that some of the same transcription factors that control the differentiation of CD8^+^ vs CD4^+^ memory may also regulate the development of different memory-like iNKT cell lineages. For example, in our study we identified enrichment of the *Zeb2* transcription factor in KLRG1^+^ iNKT cells (Figure 6B), which is known to be a key regulator of the terminal differentiation of memory CD8^+^ effector T cells^5,11^. Further research will be required to elucidate the dynamics of memory-like iNKT cell differentiation in more detail.

The question of whether TCR signal strength controls cMAF expression in iNKT cells is a fascinating one. We present some new data suggesting that increased antigen load might regulate cMAF expression under some circumstances. We performed immunization of mice with 1μg, 2.5μg or 5μg of αGalCer, which showed a dose dependent trend whereby there was an increased frequency of cMAF^+^ iNKT cells in the spleen in response to increased antigen load (Figure 6—figure supplement 2). Moreover, immuniztion of mice with 10μg of αGalCer resulted in an increased frequency of cMAF^+^ iNKT cells in the spleen (~37%) versus 4μg of αGalCer (29%), which was the dose that we had used throughout the study (Figure 5—figure supplement 4). These data suggest that cMAF expression is increased in response to increasing TCR signaling strength. Notably, expression of cMAF has previously been linked to TCR signal strength in γδ T cells^12^.

Reviewer #2 (Recommendations for the authors):1. Please add reference from the Bendelac lab re: parabiosis and tissue residence. J Exp Med. 2011 Jun 6;208(6):1179-88.

We thank the reviewer for this recommendation, and have added the reference to the manuscript.

2. In Figure 1C, the authors noted that some activated iNKT cells express genes of the KLF2 regulon and suggest that activated iNKT cells may traffic to other tissues. This should be contrasted to the work of the Kubes and Littman labs, which showed that activated iNKT cells stop patrolling blood vessels.

We thank the reviewer for highlighting the potential role of KLF2 in activated iNKT cells. We have added some discussion about the potential role of KLF2 in the Results section 1 (“iNKT cells undergo rapid and extensive transcriptional remodeling in response to αGalCer”) and contrasted our interpretation of increased KLF2 expression in activated iNKT cells versus previous work from the Kubes and Littman labs.

3. In Figure 1D, the analysis of metabolic pathway is interesting. However, I would consider toning down some of the conclusions at the data is limited to gene expression and pathway analyses. Functional metabolic studies are necessary to draw firm conclusions.

We have adjusted the language used in our manuscript and toned down our conclusions from these data, emphasizing that our analysis in Figure 1D is at the gene/pathway level.

4. The experimental setting in Figure 2H should be detailed. Were subsets sorted prior to in vitro stimulation, or were total thymic cells stimulated? Were iNKT subsets discriminated by FACS using transcription factor staining on top of cytokine antibodies. A similar experiment should be done with cells from the spleen. This experiment used Balb/c mice. Was the in vivo activation performed earlier in the manuscript performed with Balb/c or C57Bl/6 mice? The background of mice used should be clarified for all experiments. In B6 mice, iNKT1 cells are known to produce IL-4 in addition to IFN-g. Is IL-4 production by iNKT1 cells also more affected by oligomycin than their production of IFN-g?

We thank the reviewer for highlighting this so that we can further explain. The experimental system used for Figure 2H has previously been published and described by the Godfrey lab (please see Cameron *et al.* (2018), DOI: https://doi.org/10.1111/imcb.12034)^15^ as an effective model system for the study of NKT1, NKT2, and NKT17 cell subsets without the need to use transcription factors, so that later work can be performed on live cells. For example, mitochondrial staining with Mitotracker Green FM and TMRM is not possible in cells that have been fixed and permeabilized using transcription factor staining, but works well in live unfixed cells. NKT2 and NKT17 cells are also enriched in BALB/c thymus compared to many other organs in BALB/c or B6 mice^17,18^, which is also advantageous for the study of these typically rare iNKT subsets. For this reason, we employed the use of this model system to study NKT1, NKT2, and NKT17 cell function. However, we found that this staining method only works for BALB/c mice and is not clearly distinct enough in B6 mice. Briefly, functionally mature NKT1, NKT2, and NKT17 cells can be identified in the thymus of BALB/c mice using the surface markers ICOS and the activationassociated glycoform of CD43 (CD43-HG). NKT17 cells are CD43-HG^high^, NKT2 cells are ICOS^+^, and NKT1 cells are ICOS^-^. CD44 was additionally used as a marker to exclude functionally immature developing iNKT cells. We confirmed the identify of the NKT1, NKT2 and NKT17 cell subsets gated using this surface marker system using the transcription factor markers T-bet, PLZF and RORγT by flow cytometry (Figure 2—figure supplement 4). To clarify the background of mice used for all experiments in our study: BALB/c mice were only used to generate the data from Figures 2F, 2G and 2H, and Figure 2—figure supplement 4. All other experiments in our study, including scRNA-Seq experiments and in vivo stimulation experiments, were performed using C57BL/6 mice. BALB/c mice were only used for the purpose of studying NKT1, NKT2 and NKT17 cell subsets using the model system published by Cameron *et al.* (2018)^15^. To clarify the experimental procedure for the data from Figure 2H:

Total BALB/c thymocytes were stimulated with PMA (50ng/mL) and Ionomycin (1μg/mL) in complete RPMI media in the presence or absence of 40nM Oligomycin for 4 hours ex vivo before being fixed and stained for the presence of cytokines.

We have not had success using the Cameron *et al.* (2018) model^15^ with splenic iNKT cells, and therefore had to measure total cytokine production by iNKT cells without being able to gate on specific iNKT cell subsets when examining the effect of oligomycin on cytokine production by iNKT cells in the spleen. We found that incubation of total splenic iNKT cells from C57BL/6 mice with 20nM oligomycin reduced the frequency of IL-4^pos^ IFNγ^neg^ iNKT cells and IL-4^pos^ IFNγ^pos^ iNKT cells, but not IL-4^neg^ IFNγ^pos^ iNKT cells after PMA and Ionomycin stimulation (Figure 2—figure supplement 5). We did not detect a sufficiently robust IL-17A production to determine if IL-17A expression was inhibited by oligomycin in the spleen. These data suggest that oxidative metabolism is critically important for the production of IL-4 by splenic iNKT cells, including NKT1 cells co-producing IL-4 and IFNγ, but not for production of IFNγ alone by NKT1 cells. Collectively, our functional data from the thymus and spleen suggest that oxidative metabolism may be critically important for production of Th2 and Th17 cell cytokines by iNKT cells, but less so for production of the NKT1 cell cytokine IFNγ. We also performed an experiment with Mitotracker Green FM and TMRM staining in the spleen. We identified an increased frequency of NK1.1^-^ cells exhibiting higher Mitotracker Green FM and TMRM staining compared to NK1.1^+^ cells among splenic iNKT cells from C57BL/6 mice (Figure 5—figure supplement 4), indicative of increased mitochondrial mass and membrane potential in NK1.1^-^ versus NK1.1^+^ splenic iNKT cells. Since expression of NK1.1 is greatly enriched among NKT1 cells versus NKT2 and NKT17 cells^18,19^, our data suggest, as with our thymic data from BALB/c mice, that NKT2 and NKT17 cells are more biased towards mitochondrial/oxidative metabolism than NKT1 cells.

5. The authors state: "We previously found that Nur77 is enriched in adipose iNKT cells compared to splenic iNKT cells at steady state (17), but Nur77 expression does not increase upon activation in adipose iNKT cells, unlike splenic iNKT cells. Is this based on the transcriptomic data, or has this been confirmed by FACS using Nur77 antibodies, or Nur77 reporter mice?

The statement “We previously found that Nur77 is enriched in adipose iNKT cells compared to splenic iNKT cells at steady state” is based on transcriptomic microarray analysis in Lynch *et al.* (2015)^16^, flow cytometry validation of Nur77 protein expression in our recent study LaMarche et al. (2020)^3^ and transcriptomic data from our manuscript. Please see Figures S2C and S2D from LaMarche *et al.* (2020), DOI: 10.1016/j.cmet.2020.05.017, which show increased expression of Nur77 in adipose versus splenic iNKT cells at steady state. To better illustrate this in our manuscript, we have added a citation of the LaMarche *et al.* (2021) study to the statement in question.

The statement “but Nur77 expression does not increase upon activation in adipose iNKT cells, unlike splenic iNKT cells” is actually slightly incorrect and is a mistake carried over from an earlier draft of the manuscript. We thank the reviewer for helping us spot this error so that it can be corrected. *Nr4a1* expression is increased in a minority of adipose iNKT cells at 4 hours post-αGalCer. This is shown in Figure 4A and Figure 4B of the manuscript, where *Nr4a1* expression is increased in a minority of adipose NKT1 and NKT17 cells at 4 hours post-αGalCer. However, it is correct to state that *Nr4a1* expression is not increased upon activation in most adipose iNKT cells (at 4 hours post-αGalCer). We also show in Figure 3C that the magnitude of increase in *Nr4a1* expression is significantly reduced in adipose iNKT cells versus splenic iNKT cells upon activation. Therefore, we have amended the statement highlighted by the reviewer, and the entire sentence now reads:

“We previously found that Nur77 is enriched in adipose iNKT cells compared to splenic iNKT cells at steady state^17,19^, but these data suggest that adipose iNKT cells demonstrate reduced upregulation of Nur77 compared to splenic iNKT cells upon activation.”.

6. The blunted initial response observed for adipose tissue iNKT cells could rather be indicative of a different kinetic of activation between spleen and adipose tissue, due to sub-anatomical location of iNKT cells in these tissues, and accessibility to aGalCer and CD1d (see work from the Hogquist lab). A fine kinetic study of adipose tissue vs. spleen iNKT cell response may be warranted to draw this conclusion. This is in line with the statement drawn from Figure 3D-E that adipose tissue iNKT cells are more activated at baseline. Maybe they respond faster following aGalCer administration than their spleen counterparts.

We thank the reviewer for this recommendation. We are aware of the interesting work from the Hogquist lab and this is an interesting point. We did not rule out in this study that the sub-anatomical location of adipose tissue might have an influence on the activation kinetic of adipose iNKT cells in response to αGalCer. However, we have previously shown that adipose iNKT cells also display reduced production of IFNγ, a key flagship iNKT cell cytokine, compared to splenic iNKT cells following in vitro activation with αGalCer; please see Figure 1D of Lynch *et al.* (2012). Therefore, we do not believe that this blunted or delayed activation phenotype that we observe in vivo is an artifact of differential activation due to the location of adipose tissue, but rather that is indicative of a different activation phenotype between adipose and splenic iNKT cells. To further support our argument, we show that adipose iNKT cells are still actively expressing cytokine and activation marker transcripts at 72 hours post-αGalCer, whereas splenic iNKT cells have mostly ceased expressing these transcripts by 72 hours post-αGalCer (Figure 3—figure supplement 1).

7. The suggestion (from supplementary Figure 3) that iNKT10 might be functionally heterogenous is interesting and the data seems convincing. However, this could be due to subtle differences in the activation kinetic in vivo. Is there a way to fate-map some of these genes following aGalCer activation in vivo?

We thank the reviewer for this recommendation. To our knowledge it would exceedingly difficult to fate-map most of the genes from Supplementary Figure 3B in the original manuscript (now Figure S in the revised manuscript) in vivo following αGalCer treatment using something like a reporter mouse model, as this would likely require the generation of several new bespoke mouse models and likely take years. We agree that this is a cool suggestion but is outside of the scope of the current study.

8. On Figure 5A, it seems to me that steady-state and resting (4 weeks post-aGalCer) iNKTs cluster together while activated and reactivated iNKTs cluster together. The "greatly reduced response" and the similarity with activated adipose tissue iNKTs is no obvious from this UMAP data. The authors should clarify this.

We thank the reviewer for this helpful comment that we now realize needs clarification. The phrase “greatly reduced response” is in reference to Figure 5B, which shows greatly reduced expression of activation and cytokine transcripts and reduced remodeling of metabolic genes among reactivated splenic iNKT cells compared to activated splenic iNKT cells. We agree that this reduced response is not as obvious from the UMAP in Figure 5A as it is with adipose iNKT cells from the UMAP in Figure 3A. Therefore, we have made two modifications to the manuscript. We have modified the UMAP in Figure 5A by splitting the data in steady state and activated (left panel) and resting and reactivated (right panel) so that the increased overlap between resting and reactivated iNKT cells is clearer. We believe that the reduced response is shown at the gene level in Figure 5B, but the difference between activated and reactivated splenic iNKT cells is not as obvious in the UMAP as it is with adipose, we have modified our language from “greatly reduced response” to “reduced response” to be more accurate.

9. In Figure 6A, please clearly describe in the text what markers were used to define iNKT subsets among steady-state and resting I NKT cells.

We thank the reviewer for this recommendation. We have reworked the Results text relating to Figure 6A to better describe key genes and markers used to demarcate the iNKT cell subsets identified among steady state and resting splenic iNKT cells in Figure 6A. We have also added three supplemental figures (Figure 2figure supplement 1, Figure 2—figure supplement 2, Figure 6—figure supplement 1) which help describe how we identified and defined iNKT cell subsets among steady state and resting iNKT cells. These new supplemental figures are referenced in the text describing the identification of iNKT cell subsets for Figure 6A.

10. How do the clusters A-C described in Figure 6G compare with effector subsets described in Figure 4?

The cells in Figure 6G of the original manuscript are NK1.1^-^ adipose NKT1 cells at steady state. In Figure 4 we described a range of different clusters of adipose iNKT cells at steady state and at 4 hours post-αGalCer. The cells in Figure 6G of the original manuscript (now Figure 6—figure supplement 5 of the revised manuscript) are the same cells as the cells in Cluster 2 in Figure 4A (steady state adipose NK1.1^-^ NKT1 cells).

11. "Flow cytometry of splenic iNKT cells 4 weeks post-⍺GalCer confirmed that IL-10+ iNKT cells expressed cMAF, and cMAFneg cells produced little IL-10 (Figure 7F, Figure 7G)." These panels do not appear to show FACS data. Does this relate to panel D? It would be better to show IL-10 vs. cMAF from CD1d tetramer-positive cells on these plots rather than CD1d tetramer staining. The figure should also use PMA/Iono stimulation of steady-state iNKT cells from the same tissue as a control.

We thank the reviewer for pointing out this typographical error – the line "Flow cytometry of splenic iNKT cells 4 weeks post-αGalCer confirmed that IL-10^+^ iNKT cells expressed cMAF, and cMAFneg cells produced little IL-10 (Figure 7F, Figure 7G)." should have instead referred to Figure 7D and Figure 7E from the original manuscript, which referenced the flow cytometry data. This is now corrected. We also thank the reviewer for the recommendation about how to display the flow data in Figure 7D of the original manuscript. For this analysis we pre-gated on cMAF^+^ and cMAF^-^ iNKT cells before gating for expression of IL-10, in order to show that cMAF^+^ iNKT cells produce more IL-10 than cMAF^-^ iNKT cells. We have, however, amended the panel to show the data in the format suggested by the reviewer. In addition, we have also added a panel of steady state iNKT cells from the same tissue (spleen). Please note – these revised flow cytometry data showing expression of IL-10 versus cMAF have been moved to from Figure 7 to Figure 5F of the revised manuscript, based on a request from reviewer #3.

Reviewer #3 (Recommendations for the authors):This paper is of interest to scientists within the field of iNKT cells. The authors conducted scRNA-seq to longitudinally profile activated iNKT cells and generated a transcriptomic atlas of iNKT cells at the activation states. The study suggests that transcriptional signatures of activation are highly conserved among heterogeneous iNKT cell populations and that the adipose iNKT cells undergo blunted activation and display constitutive enrichment of memory like population. In addition, the study also identifies a conserved cMAF- associated network in NKT10 cells. The data quality is relatively high.1. Based on the violin plots shown in Figure 2B, in the 4 hours aGalCer stimulation group, it seems that Tbx21 and Zbtb16 are evenly spaced in all three clusters, where it is difficult to identify NKT1, NKT2 in this case. It would be clearer if the author could show NKT_1/2_/17 signatures expression pattern in umap plot.

We thank the reviewer for this recommendation. This violin plot is correct, as we found that expression of *Tbx21* (T-bet) and *Zbtb16* (PLZF) did not effectively demarcate NKT1 and NKT2 cells at the RNA level after αGalCer, unlike at steady state, due to activation induced changes (see the left-hand side of Figure 2B). Therefore, we instead used flagship NKT1, NKT2, and NKT17 cell cytokines (*IFNγ*, *Il4*, *Il13*, *Il17a* and *Il17f*) to demarcate NKT1, NKT2, and NKT17 cell subsets. We have prepared an additional two supplemental figures (Figure 2—figure supplement 1 and Figure 2—figure supplement 2) and added some text to the Methods (see ”Downstream scRNA-Seq data analysis”) which should better clarify how we identified NKT1, NKT2 and NKT17 cells. These new supplemental figures also include UMAP plots of NKT1, NKT2 and NKT17 cell signature gene expression in accordance with the useful recommendation from the reviewer.

2. The statement " Notably, we found that all subsets upregulated Zbtb16 and Tbx21 after activation (Figure 2B), suggesting that PLZF and T-bet play subset-independent roles during activation." This needs more evidence. The gene expression level (such as Tbx21, Zbtb16) increased post aGalCer activation could be due to the transcriptional bursting, author should show and compare the house keeping genes expression level in both steady state and aGalCer activation state.

We thank the reviewer for this comment and recommendation. Increased PLZF expression after iNKT cell activation has previously been shown by Oleinka *et al.* (2018)^20^, (DOI:10.1038/s41467-018-02911-y), which is in line with what we found. However, increased expression of T-bet (*Tbx21*) has not previously been demonstrated among activated iNKT cells. We agree with the reviewer that more evidence is needed to show subset independent roles and so we have amended the referenced statement in our manuscript to “Notably, we found that all subsets upregulated *Zbtb16* after activation (Figure 2B), suggesting that PLZF may play a subset-independent role during activation.”. For *Tbx21*, we clarify instead that *Tbx21* expression is non-specifically increased across all iNKT cell subsets after activation at the RNA level, and for this reason we did not use *Tbx21* as a marker gene for iNKT cell subset identification after activation.

3. In Figure 2D, using correlation heatmaps the authors mapped the gene profiles in different subsets of NKT cells before and after aGalCer stimulation, and claimed that "activated NKT2 and NKT17 cells are transcriptionally similar compared to NKT1 cells." The authors should consider that post aGalCer stimulation, proliferating genes/cells will be predominant. The similarity of NKT2 and NKT17 observed in the heatmap could be due to the overwhelming cell cycle genes, instead of the intrinsic genes of NKT2 and NKT17. The authors should remove cell cycle gene, and then do the mapping.

We thank the reviewer for highlighting this point. We demonstrate that cell cycle and proliferation genes are not strongly upregulated by 4 hours post-αGalCer (see Figure 1C, Figure 3D, and a new Figure 1—figure supplement 1). There is a strong upregulation of cell cycle and proliferation genes by 72 hours post-αGalCer (Figure 1, Figure 3), and as the reviewer points out this could dominate the any results including data from 72 hours post-αGalCer. For that reason we had performed cell cycle regression in all of our cross-analysis of scRNA-Seq data from 72 hours post-αGalCer with scRNA-Seq data from different time points, such as steady state or 4 hours post-αGalCer (see Figure 1 and Figure 3 for example). Therefore, we believe that it is unnecessary to perform cell cycle regression for the correlation analysis of iNKT cells at 4 hours post-αGalCer in Figure 2D, and we believed that doing so might lead to artifacts from an unnecessary gene regression being applied, similar to how performing batch correction on a scRNA-Seq dataset without a batch effect can lead to strange artifacts appearing in the data.

4. The study claimed that NKT10 and Tr1 share transcriptional features. It will make this study more interesting if the author could employ the experiments to check on their function potential.

We agree that this is a very interesting idea and could make for a cool follow-up study. However, in our study we only referenced Tr1 cells with respect to some overlap of gene signatures between what we found for adipose iNKT cells, splenic iNKT cells at 4 weeks post-αGalCer and what has been published in the literature for Tr1 cells. Therefore, we respectfully suggest that a detailed comparison of adipose iNKT cells and Tr1 cells on a functional level is outside of the scope of this study.

5. Figure 2F only shows the representation of the experiment, statistics summary bar graphs should also be included.

We agree, and we have added statistics summary bar graphs for this experiment in Figure 2G as requested, replacing the scatter plot present in the original manuscript.

6. Figure 3B is generated from bulk analysis of scRNA-Seq data, right? What about by Stages (4 and 72h) subclusters?

We apologize for any confusion with Figure 3B, we found this slightly more difficult to explain in a short space. To generate the analysis in Figure 3B we performed cross-dataset differential gene expression analysis of adipose and splenic iNKT cells. This was performed by first individually comparing adipose versus splenic iNKT cells at steady state, 4 hours post-αGalCer and 72 hours post-αGalCer using gene expression analysis, generating three different lists of differentially expressed genes between adipose and splenic iNKT cells, one list for each time point or comparison. We then identified genes which were significantly enriched in either adipose or splenic iNKT cells across all three comparisons. This identified a total of 971 genes enriched among adipose iNKT cells and a total of 65 genes enriched among splenic iNKT cells. We then performed overrepresentation analysis of these enriched genes using gprofiler^21^ to generate the pathway analysis shown in Figure 3B. To clarify this in the manuscript we have added additional explanation to the Methods (see “Downstream scRNA-Seq data analysis”) and referenced this in the text to aid the reader in understanding our analysis. The reason that we did our analysis in this way, rather than just performing a bulk comparison of all adipose and splenic iNKT cells at steady state, 4 hours post-αGalCer and 72 hours post-αGalCer together, is that our method allows us to better leverage the different temporal data that we have access to in our scRNA-Seq data, and enables the identification of genes which are always (no matter what activation state) enriched among adipose or splenic iNKT cells, providing insight into core genes associated with the phenotype of iNKT cells in the adipose tissue or spleen. We hope that this explanation provides more clarity for the reader.

7. "Analysis of cytokine production among adipose iNKT cells revealed that IL-10 was only expressed by NKT1 cells (Figure 4B/D)." Better have flow data to support this claim. No scale stick in Y axis in Figure 4C.

We note that the referenced statement in our manuscript reads “*Il10* was only expressed by NKT1 cells”, which refers to *Il10* gene expression being specific to NKT1 cells. We show this by heatmap in Figure 4B. However, we found that IL-10 and IFNγ were co-produced by splenic and adipose iNKT cells at 4 weeks post-αGalCer (Figure 6—figure supplement 3), and IL-10 and IFNγ were also co-produced by splenic iNKT cells at 72 hours post-αGalCer (Figure 1—figure supplement 3). As IFNγ is a flagship NKT1 cell cytokine, these data suggest that iNKT10 cells are an NKT1-like population (or were previously NKT1 cells). We have added scale sticks to the Y axes for Figure 4C.

8. Clusters showing in Figure 5F with NKT 10 cells are better validated in protein level by FACS.

We thank the reviewer for this recommendation and agree. We have reworked Figure 5, reducing the number of scRNA-Seq plots present (see Figure 5E), and we present new data showing co-expression of KLRG1 and Granzyme A (see Figure 5F) by flow cytometry, to better illustrate the identification of functional KLRG1^+^ iNKT cells among reactivated splenic iNKT cells. We have also reworked and moved flow cytometry data from Figure 7 of the original manuscript to now show co-expression of cMAF and IL-10 in the reworked Figure 5 (see Figure 5G), to better illustrate the identification of functional cMAF^+^ iNKT cells among reactivated splenic iNKT cells. We have also reworked the Results text associated with Figure 5 to reflect these new data.

9. Clusters showing in Figure 6G should be validated in protein level.

We thank the reviewer for this recommendation. The goal of Figure 6G in the original manuscript was to indicate that iNKT cell populations similar to the memory-like KLRG1^+^ and cMAF^+^ iNKT cells that we identified by scRNA-Seq and validated by flow cytometry in the spleen after prior αGalCer immunization (Figure 6C and 6D of the original manuscript) are always detected in the adipose tissue at steady state. In our analysis of adipose NK1.1^-^ iNKT cells in Figure 6G of the original manuscript, we identified Cluster C, which might be representative of KLRG1^+^ iNKT cells present at steady state in the adipose tissue, and Cluster B, which might be representative of a cMAF^+^ iNKT cell population present in adipose tissue at steady state. We have now reworked Figure 6 and included new FACS data showing distinct KLRG1^+^ iNKT cell and cMAF^+^ iNKT cell populations present in the adipose tissue at steady state and after αGalCer treatment (see Figure 6D-6F). These new FACS data make the same point as the previous scRNA-Seq data did about KLRG1^+^ and cMAF^+^ iNKT cells being present in the adipose tissue at steady state while additionally showing the enrichment of KLRG1^+^ and cMAF^+^ iNKT cells across multiple organs after αGalCer, which was a request from reviewer #1. The scRNA-Seq data from Figure 6E, 6F, 6G and 6H of the original manuscript are now present in Figure 6—figure supplement 5. We have also reworked the Results text associated with Figure 6 to reflect these new data.

10. The author did not mention how many biological replicates for scRNA-seq experiment, since mice treated with activator could vary widely between mouse to mouse, at least 2 biological replicates are needed.

Thank you to the reviewer for highlighting this. In the “scRNASeq sequencing and data pre-processing” section of the Methods we state that “A total of thirty-five visceral adipose tissue deposits and five spleens were pooled for each sample”. To be more precise on this point, we have amended this statement to say “A total of thirty-five visceral adipose tissue deposits from 35 mice or five spleens from 5 mice were pooled for each scRNA-Seq sample”.

11. The author mentioned in the method that both C57BL6 mice and BALB/c mice were used in this study, but did not mention which experiments were performed using C57BL6 mice or BALB/c. NKT cell subsets are different between C57BL6 and BALB/C mice. Please discuss the rationale why these two strains were used and make them clearer.

We agree that it is confusing without further clarification, as reviewer 2 also mentioned, and we have added a clarification to the “Animals” section of the Methods stating that “C57BL/6 mice were used for all experiments unless otherwise specified” and have made it more clear in both the Figure Legends and Results text associated with Figure 2 and Figure 2—figure supplement 4, which are the only figures where BALB/c mice were used in some experiments. C57BL/6 mice were used for almost all experiments in our study, including the generation of all scRNA-Seq data. BALB/c mice were specifically used for the purpose of investigating the phenotype and function of NKT1, NKT2 and NKT17 cells using an excellent thymic model previously described in a publication by Cameron *et al.* (2018)^15^ from the Godfrey lab, in which functionally NKT1, NKT2 and NKT17 cells can be easily labeled using the surface markers ICOS and the activation-associated glycoform of CD43 (CD43-HG), so that the cells do not need to be fixed and permeabilized. This allows us to perform later assays on live cells, such as mitochondrial analysis, which is not possible if fixation and permeabilization buffer is used for transcription factor analysis. However, we found that this method only works in BALB/c mice and not well in C57BL/6 mice, hence for Figure 2 and Figure 2—figure supplement 4 only, BALB/c mice were used.

References:

1. Shimizu, K. et al. KLRG+ invariant natural killer T cells are long-lived effectors. Proc. Natl. Acad. Sci. U. S. A. 111, 12474–12479 (2014).

2. Murray, M. P. et al. Transcriptome and chromatin landscape of iNKT cells are shaped by subset differentiation and antigen exposure. Nat. Commun. 12, 1–14 (2021).

3. LaMarche, N. M. et al. Distinct iNKT Cell Populations Use IFNγ or ER Stress-Induced IL-10 to Control Adipose Tissue Homeostasis. Cell Metab. 32, 243-258.e6 (2020).

4. Sag, D., Krause, P., Hedrick, C. C., Kronenberg, M. and Wingender, G. IL-10–producing NKT10 cells are a distinct regulatory invariant NKT cell subset. J. Clin. Invest. 124, 3725–3740 (2014).

5. Omilusik, K. D. et al. Transcriptional repressor ZEB2 promotes terminal differentiation of CD8+ effector and memory T cell populations during infection. J. Exp. Med. 212, 2027– 2039 (2015).

6. Obar, J. J. et al. Pathogen-Induced Inflammatory Environment Controls Effector and Memory CD8 + T Cell Differentiation. J. Immunol. 187, 4967–4978 (2011).

7. Joshi, N. S. et al. Inflammation Directs Memory Precursor and Short-Lived Effector CD8(+) T Cell Fates via the Graded Expression of T-bet Transcription Factor. Immunity 27, 281 (2007).

8. Plumlee, C. R. et al. Early Effector CD8 T Cells Display Plasticity in Populating the Short-Lived Effector and Memory-Precursor Pools Following Bacterial or Viral Infection. Sci. Rep. 5, 1–13 (2015).

9. Kallies, A., Zehn, D. and Utzschneider, D. T. Precursor exhausted T cells: key to successful immunotherapy? Nat. Rev. Immunol. 2019 202 20, 128–136 (2019).

10. Ricardo Miragaia, A. J. et al. Single-Cell Transcriptomics of Regulatory T Cells Reveals Trajectories of Tissue Adaptation. (2019) doi:10.1016/j.immuni.2019.01.001.

11. Evrard, M. et al. Sphingosine 1-phosphate receptor 5 (S1PR5) regulates the peripheral retention of tissue-resident lymphocytes. J. Exp. Med. 219, (2021).

12. Zuberbuehler, M. K. et al. The transcription factor c-Maf is essential for the commitment of IL-17-producing γδ T cells. Nat. Immunol. 20, 73–85 (2019).

13. Fujino, M. et al. c-MAF deletion in adult C57BL/6J mice induces cataract formation and abnormaldifferentiation of lens fiber cells. Exp. Anim. 69, 242 (2020).

14. Gabryšová, L. et al. C-Maf controls immune responses by regulating disease-specific gene networks and repressing IL-2 in CD4+ T cells article. Nat. Immunol. 19, 497–507 (2018).

15. Cameron, G. and Godfrey, D. I. Differential surface phenotype and context-dependent reactivity of functionally diverse NKT cells. Immunol. Cell Biol. 96, 759–771 (2018).

16. Lynch, L. et al. Regulatory iNKT cells lack expression of the transcription factor PLZF and control the homeostasis of Treg cells and macrophages in adipose tissue. Nat. Immunol. 16, 85–95 (2015).

17. Lee, Y. J. et al. Tissue-Specific Distribution of iNKT Cells Impacts Their Cytokine Response. Immunity 43, 566–578 (2015).

18. Lee, Y. J., Holzapfel, K. L., Zhu, J., Jameson, S. C. and Hogquist, K. A. Steady-state production of IL-4 modulates immunity in mouse strains and is determined by lineage diversity of iNKT cells. Nat. Immunol. 14, 1146–1154 (2013).

19. Engel, I. et al. Innate-like functions of natural killer T cell subsets result from highly divergent gene programs. Nat. Immunol. 17, 728–39 (2016).

20. Oleinika, K. et al. CD1d-dependent immune suppression mediated by regulatory B cells through modulations of iNKT cells. Nat. Commun. 9, 1–17 (2018).

21. Raudvere, U. et al. G:Profiler: A web server for functional enrichment analysis and conversions of gene lists (2019 update). Nucleic Acids Res. 47, W191–W198 (2019).